# Large T cell clones expressing immune checkpoints increase during multiple myeloma evolution and predict treatment resistance

Tumor recognition by T cells is essential for antitumor immunity. A comprehensive characterization of T cell diversity may be key to understanding the success of immunomodulatory drugs and failure of PD-1 blockade in tumors such as multiple myeloma (MM). Here, we use single-cell RNA and T cell receptor sequencing to characterize bone marrow T cells from healthy adults ($n = 4$) and patients with precursor ($n = 8$) and full-blown MM ($n = 10$). Large T cell clones from patients with MM expressed multiple immune checkpoints, suggesting a potentially dysfunctional phenotype. Dual targeting of PD-1 + LAG3 or PD-1 + TIGIT partially restored their function in mice with MM. We identify phenotypic hallmarks of large intratumoral T cell clones, and demonstrate that the CD27⁻ and CD27⁺ T cell ratio, measured by flow cytometry, may serve as a surrogate of clonal T cell expansions and an independent prognostic factor in 543 patients with MM treated with lenalidomide-based treatment combinations.

T cells mediate antitumour immunity after the recognition of antigenic peptides, presented by the major histocompatibility complex (MHC) or human leukocyte antigen (HLA) class I molecules on tumor cells[1,2]. Accordingly, tumor-infiltrating lymphocytes (TILs) are central players in the tumor microenvironment (TME), shaping fundamental clinical properties such as progression from benign to malignant states and response to immunotherapies[3,4]. Indeed, re-activation and clonal expansion of tumor-reactive T cells are critical to the success of immune checkpoint blockade (ICB), adoptive transfer of TILs and immunomodulatory drugs (IMIDs)[5].

Host immune surveillance mediated by tumor-reactive T cells and some immunotherapy drugs rely in the interaction of unique T cell receptors (TCR) with cognate peptide-MHC molecules[6]. Increasing evidence indicate that only a proportion of TILs are able to recognize tumor antigens[7]. Thus, the lack of intrinsic tumor reactivity in a large fraction of intratumoral T cells implies that studies examining T cell differentiation and exhaustion states in the tumor microenvironment (TME) will essentially be assessing the phenotypic state of a large

number of bystander T cells that are irrelevant to tumor control[8]. Therefore, knowledge about the phenotype of tumor-reactive T cells is a prerequisite for informative immune monitoring of malignant transformation and immunotherapy response[5].

Despite the extensive use of immunotherapies in hematological malignancies, the single-cell landscape of TILs in these patients lags behind what has been accomplished in solid tumors. One such example is multiple myeloma (MM), an incurable plasma cell malignancy for which treatment is being redefined by immunotherapies[9], where single-cell studies of TILs are scarce[10–12] and the phenotype of clonotypic T cells remains unknown. MM is an exceptional model to study intratumoral T cells because of three singularities. First, it progresses from well-defined premalignant conditions through different immune escape mechanisms[13] and immunotherapies are being investigated to prevent malignant transformation[14,15]. Second, IMIDs are a backbone of MM treatment, but there are no T cell markers to predict clinical benefit. Third, other immunotherapies relying on the

e-mail: cirino.botta@unipa.it; bpaiva@unav.es

activity of MM-clonal T cells such as PD-1 blockade failed to prolong patient's survival[16,17].

Here, we aimed to define the phenotype of individual T cell clones throughout myelomagenesis. To this end, we performed single-cell RNA and TCR sequencing of bone marrow (BM) T cells from patients with newly diagnosed MM and its precursor states, monoclonal gammopathy of undetermined significance (MGUS) and smoldering multiple myeloma (SMM). We identify large clonotypic expansions characterized by the expression of distinct combinations of immune checkpoint molecules. We also evaluate the effect of ICB treatments corresponding to each of these phenotypes in experimental MM models. Lastly, we identify T cell phenotypes that are predictive of survival in patients who received lenalidomide-based treatment combinations.

## Results

### The T cell compartment in healthy, benign, and malignant bone marrow

We used fluorescence activated cell sorting (FACS) to isolate BM T cells from four healthy adults, eight cases with the precursor states, MGUS and SMM ($n = 4$ each), and ten patients with active, newly-diagnosed MM (Fig. 1A and Table S1). Droplet-based 5′ scRNA-seq and paired scTCR-seq was performed to analyze the clonal relationship and functional states among BM T cells. After stringent quality control filtering, scTCR-seq yielded one or more complementarity-determining region 3 (CDR3) of both α and β chains in 41,018 T cells with paired transcriptomes.

Because there is limited data on the effect of tumor progression in global T cell phenotypes determined by scRNA-seq[10,11,18], we first mapped BM T cell states across healthy adults, MGUS/SMM and MM patients. Overall, 63 million mRNA transcripts in 41,018 T cells from the 22 subjects described above, were sequenced. Transcriptional profiles of individual T cells allowed grouping of similar cells into clusters, which were named according to the expression levels of well-known genes (Fig. S1 and Supplementary Dataset 1). Thus, 16 subsets were identified including γδ-like T cells, double negative and double positive T cells, T regulatory (Treg) cells, five CD4+ T cell clusters (naïve, stem cell memory [SCM], central memory [CM], effector memory [EM], effector granzyme K [GZMK+] CD4+ T cells), and seven CD8+ T cell subsets (naïve, CM, effector GZMK+, EM GZMK+, GZMK+ perforin 1 [PRF1+], granzyme B [GZMB+] cells, plus an additional subset of potentially exhausted CD8+ T cells with increased expression of TOX) (Fig. 1B).

Because these T cell subsets are similar to those expected to be circulating in the peripheral blood, we investigated the quality of the aspirates based on the percentages of BM specific cell types such as B-cell precursors, mast cells and nucleated red blood cells, determined by multiparameter flow cytometry (MFC) whenever this method was performed simultaneously to scRNA- and scTCR-seq (i.e., in 5 MGUS/SMM and 7 MM patients). The three cell types were systematically detected and their percentages were within the quartile percentages observed in a previously reported reference dataset[19] (Table S2). These results preclude the risk of severe hemodilution in the BM aspirates analyzed in this study.

All 16 clusters except CD8+ TOX+ T cells were present in healthy adults, MGUS/SMM and MM patients (Fig. 1C and Table S3). The subset of potentially exhausted CD8+ TOX+ T cells was detected exclusively in the BM of 4 of the 10 MM patients (Fig. S2). Only four of the 16 clusters showed altered distribution across normal, benign and malignant BM. Namely, MM patients displayed increased percentages of Treg, CD8+ CM cells, and CD8+ effector GZMK+, together with a reduced frequency of CD8+ GZMB+ T cells (Fig. 1C and Table S3).

### The phenotype of clonal T cells during disease progression

A total of 33,243 distinct clonotypes defined by unique TCRα and TCRβ genes sequences were identified amongst the 41,018 T cells from the

22 subjects described above (Fig. 1D). Based on the percentage of each clonotype within total T cells, these were categorized as small (range, 0−<0.01), medium (range, 0.01−≤ 0.1) and large (range, 0.1−≤ 1). Overall, most clonotypes were small (average of 69% within total T cells); medium and large expanded T cell clones were less frequent (averages of 12% and 19% within total T cells, respectively). Of note, the relative distribution of small, medium and large T cell clones was similar across healthy adults, MGUS/SMM and MM (Fig. 1E, F). Accordingly, there were no significant differences in the Shannon's and inverse Simpson indexes (Fig. S3). By contrast, healthy adults showed significantly higher TCR diversity when compared to MGUS/SMM and MM patients (Fig. S3), as measured by the index scores of Chao1 (338, 187, and 185, respectively; $p = 0.014$) and abundance-based coverage estimator (407, 193, and 196, respectively; $p = 0.022$).

Elicited by these findings, we next analyzed the transcriptional phenotype of small, medium and large T cell clones in BM aspirates from healthy adults, MGUS/SMM and MM patients (Fig. 2A, Supplementary Dataset 2). Small clones were predominant in the various CD4+ clusters including Tregs, as well as in double negative and double positive T cells. By contrast, medium and large clones were seen at greater percentages in γδ-like T cells and all CD8+ clusters except for the naïve subset (Fig. 2A and Supplementary Dataset 2). These results are consistent with the fact that, amongst large clones, the ratio between CD4+ and CD8+ T cells was 1:1.4. Interestingly, the ratio between CD4+ and CD8+ large T cell clones progressively increased in BM aspirates of healthy adults, MGUS/SMM and MM patients (1:1.2, 1:1.7 and 1:2.1, respectively; Fig. S4).

We then compared the distribution of T cell clusters among small, medium and large clones (Fig. 2B, Supplementary Dataset 3). Significant differences between normal, benign and malignant BM aspirates were observed in medium and large clones. In healthy adults, these were enriched in CD8+ GZMB+ T cells ($p < 0.05$), followed by CD8+ EM GZMK+, γδ-like and CD8+ effector GZMK+ T cells. In MGUS/SMM, there was a predominance of CD8+ EM GZMK+ ($p < 0.05$) and CD8+ GZMB+ T cells. By contrast, large clones from MM patients were enriched in CD8+ GZMK+ PRF1+ ($p < 0.01$), CD8+ effector GZMK+ ($p < 0.05$) and CD8+ TOX+ ($p < 0.01$) T cells. Indeed, potentially exhausted CD8+ TOX+ T cells, which were uniquely detected in the BM of MM patients, were particularly enriched in large clones (Fig. 2A and Supplementary Dataset 2). Collectively, these results suggest that albeit similar expansion of T cell clones in the BM of healthy adults, MGUS/SMM and MM patients, there is reduced TCR diversity in benign and malignant bone marrow, and a potentially increased dysfunction of large T cell clones in MM. Accordingly, these showed increased expression of PD1 and TIGIT when compared to large T cell clones from healthy adults and MGUS/SMM patients (Fig. S5).

### ICB combination therapy tailored to the phenotype of large T cell clones

To corroborate the potential dysfunction of large T cell clones in MM, we investigated the effect of ICB in a genetically engineered $BI_{cy1}$ mouse model that results from transgenic BCL2 and IKK2[NF-κB] expression in mature germinal center B lymphocytes by the cγ1-cre allele[20] (Fig. S6). Upon T cell driven immunization with sheep red blood cells, mice spontaneously develop MM fulfilling two important characteristics of human disease: the evolution of pre-malignant MGUS into full-blown MM, and the interplay between tumor and the BM immune microenvironment during progression (Fig. S6)[20]. A total of 16,043 BM T cells from control, MGUS and MM bearing mice were characterized by scRNA-seq and scTCR-seq (Fig. 3A and Table S4). MGUS was defined as <10% GFP+CD138+B220-sIgM- BM plasma cells and no CRAB-like features (hypercalcemia, renal disease, anemia, and bone disease), whereas MM was diagnosed when mice presented >10% tumor cells and/or CRAB (Fig. S6). Similar to humans, we observed few differences in the distribution of T cell clusters in control vs MGUS and MM mice,

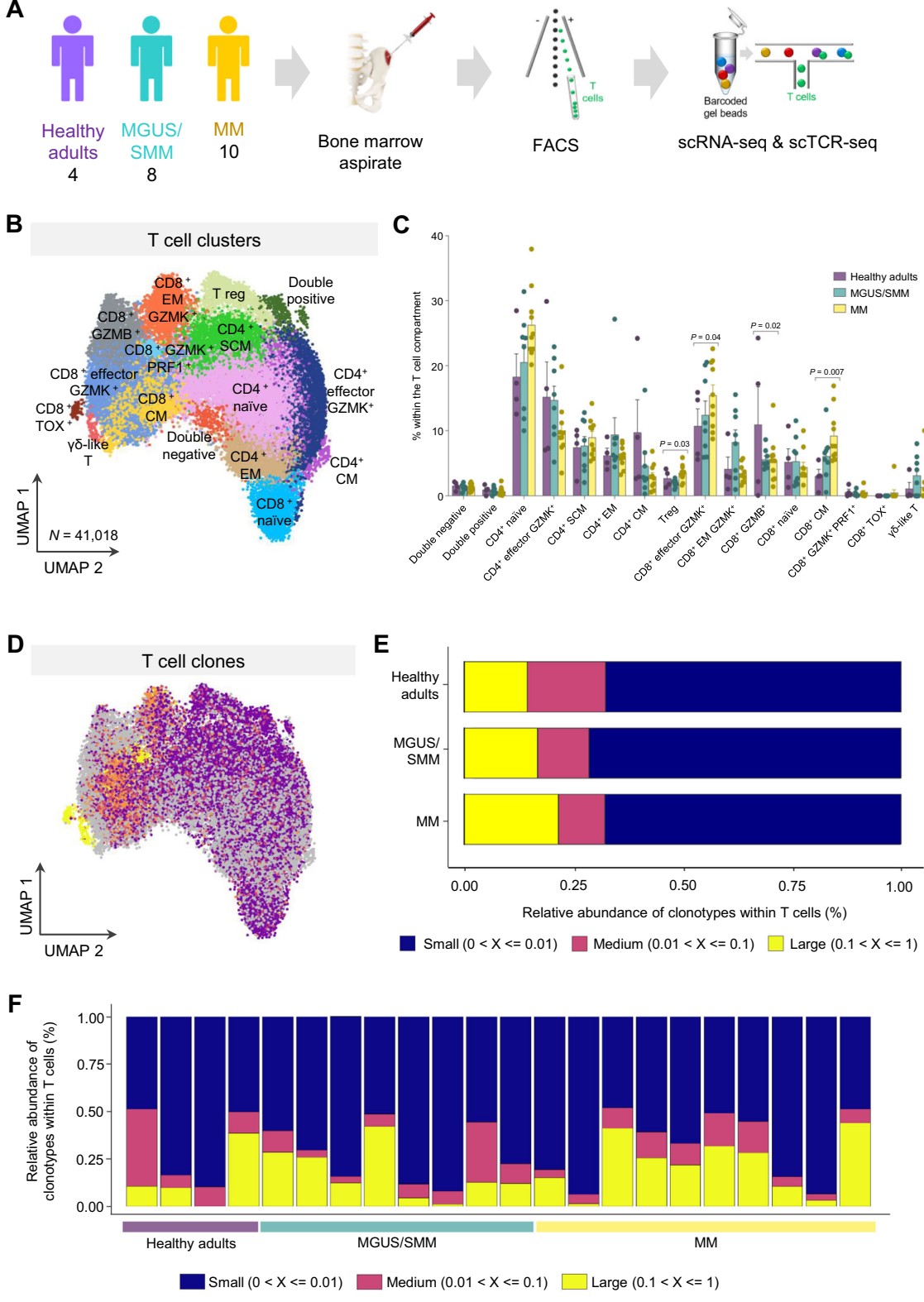

one of which being once again the increased percentage of CD8+ GZMK+ T cells in the later (Fig. 3B and Table S5).

Another similarity with humans was the different cluster distribution within large T cell clones from control vs MGUS and MM mice. The predominant cluster within medium and large clones from control mice was composed of CD8+ GZMB+ LAG3+ ($p < 0.05$) and CD8+ naïve ($p < 0.01$) T cells, whereas in MM bearing mice it was composed of CD8+ GZMK+ T cells ($p < 0.05$) (Fig. 3C,

Supplementary Dataset 4). Furthermore, large clones from MM mice showed increased expression of *CD8*, *CD38*, *Pd1* and *Tigit* when compared to large clones from control and MGUS mice (Fig. 3D). Collectively, tumor progression in immunocompetent mice that spontaneously develop MM showed similarities with human cancer, and could therefore be used to investigate if the possible dysfunction of large T cell clones in MM could be restored using ICB.

**Fig. 1 | The T cell compartment in healthy, benign, and malignant bone marrow. A** Experimental design. Bone marrow aspirates were collected from four healthy adults, eight patients with the precursor states of monoclonal gammopathy of undetermined significance (MGUS) or smoldering multiple myeloma (SMM), and ten patients with active, newly-diagnosed multiple myeloma (MM). T cells were isolated using fluorescence activated cell sorting (FACS), followed by simultaneous, single-cell sequencing of RNA (scRNA-seq) and T cell receptors (scTCR-seq). **B** Uniform manifold approximation and projection (UMAP) of 16 T cell clusters identified with single-cell RNA sequencing, in bone marrow aspirates from four healthy adults, eight patients with MGUS/SMM, and ten patients with active, newly-diagnosed MM. **C** Relative distribution of the 16 clusters within the T cell

compartment of healthy adults ($n = 4$), MGUS/SMM ($n = 8$) and MM ($n = 10$) patients. Error bars represent mean ± standard error mean (SEM). *P* values were calculated using the Kruskal–Wallis test, *$p = 0.03$, 0.04 and 0.02, **$p = 0.007$ from left to right. Source data are provided as a Source Data file. **D** Uniform manifold approximation and projection (UMAP) of the distribution of T cell clones in bone marrow T cells from healthy adults, MGUS/SMM and MM patients. **E** Bar chart showing clonal distribution in grouped healthy adults, MGUS/SMM and MM patients. T cell clones were categorized as small (range, 0–≤ 0.01), medium (range, 0.01–≤ 0.1) and large (range, 0.1–≤ 1) based on the percentage of each clonotype within total T cells. Source data are provided as a Source Data file. **F** Bar chart showing clonal distribution in individual cases.

Because large T cell clones showed co-expression of multiple immune checkpoints, the combined administration of anti-PD1 plus anti-LAG3 or anti-TIGIT was tested in immunocompetent C57BL/6 mice intravenously injected with the MM5080 murine MM cell line, which was established from BIcγ1 mice with an additional P53 deletion (Fig. S7). This model is characterized by the progressive accumulation of tumor cells in the BM, together with increased percentages of CD8⁺ T cells overexpressing PD1, LAG3 and TIGIT (Fig. S7). Mice were treated with ICB (single-agent or dual combinations) at days +3, +10 and +17. Interestingly, none of the ICB used in monotherapy prolonged survival; by contrast, the co-administration of anti-PD1 plus anti-LAG3, or anti-PD1 plus anti-TIGIT, resulted in longer overall survival (OS) (Fig. 3E). Thus, the blockade of two immune checkpoints significantly delayed MM growth.

### T cell markers of clonality and prognosis in MM

We then aimed at identifying phenotypic hallmarks of large T cell clones that could be leveraged for MFC immune monitoring. Among the 656 deregulated genes between small, medium and large T cell clones from MGUS/SMM and MM patients (Supplementary Dataset 5), 35 coded for cell surface proteins. Namely, small clones showed decreased expression of *CD3, CD8, CD16, CD48, CD52, CD53 CD63, CD74, CD99, CD247, CD320, CXCR3*, and *LAG3*, as well as higher mRNA levels of *CD27, CD28, CD55, CD62L, CD127, CCR6*, and *CCR7* (Fig. 4A and Supplementary Dataset 5).

One of the differentially expressed antigens across small, medium and large T cell clones was *CD27*; the latter showing significantly lower mRNA levels (Fig. 4B, C). This was an important observation because CD27 is amongst the markers that are routinely evaluated through MFC for the screening of plasma cell clonality in patients with monoclonal gammopathies[21,22]. Of note, there was a significant correlation between the number of T cell clones and the ratio between CD27⁻ and CD27⁺ T cells (CD27⁻:CD27⁺), as determined by scRNA-seq and MFC whenever these were simultaneously performed in BM aspirates from MGUS/SMM and MM patients (Fig. 4D, E). Thus, the extent of clonal T cell expansions could be estimated in larger series of patients with available immunophenotypic data, according to the CD27⁻:CD27⁺ ratio.

To investigate its prognostic value, we analyzed 271 and 272 MM patients that were respectively enrolled in the transplant-ineligible GEM-CLARIDEX[23] and transplant-eligible GEM2012MENOS65[24] phase 3 clinical trials. Both series yielded unique opportunities to investigate the prognostic value of the CD27⁻:CD27⁺ ratio as a surrogate of clonal T cell expansions in patients treated with regimens including lenalidomide, an IMID that requires tumor-reactive T cells to mediate its anti-MM effect[25].

Using computational flow cytometry, we identified 22 BM clusters including CD27⁻ and CD27⁺ T cell subsets (Fig. 5A and Fig. S8). The presence of a CD27⁻:CD27⁺ ratio higher than the median value (i.e., ≥0.3) was significantly associated with longer progression-free survival (PFS) in both transplant-ineligible (hazard ratio: 0..597, 95% confidence interval: 0.366–0.975; $p = 0.0402$) (Fig. 5B) and transplant-eligible

patients (hazard ratio: 0.493, 95% confidence interval: 0.289–0.840; $p = 0.0052$) (Fig. 5C). The presence of a CD27⁻:CD27⁺ ratio ≥0.3 was significantly associated with longer OS in transplant-eligible patients (Fig. S9). Furthermore, multivariate analysis including the prognostic factors that define the revised International Staging System (i.e., ISS, lactate dehydrogenase [LDH] and cytogenetic risk) demonstrated that the CD27⁻:CD27⁺ ratio showed independent prognostic value for PFS (Fig. 5D). The CD27⁻:CD27⁺ ratio was not associated with patients' staging and LDH levels (Table S6), nor the presence of cytogenetic abnormalities such as +1q, del(1p), del(17p) and IgH chromosomal translocations (Table S7). Furthermore, there were no differences in the mutational burden and transcriptional profile of tumor cells, respectively assessed by whole-exome sequencing ($n = 23$) and RNA-seq ($n = 102$), between patients with <0.3 and ≥0.3 CD27⁻:CD27⁺ratio (Fig. S10).

To gain further insight into a putative association between clonal T cell expansions in the TME (i.e., high CD27⁻:CD27⁺ ratio) and their re-activation upon exposure to lenalidomide, we cultured whole-BM aspirates from MM patients ($n = 3$) in a 3D organoid for 5 days, and treated with 1 μM lenalidomide +/− an anti-HLA antibody to block TCR-MHC interactions. The significant tumor cell killing induced by lenalidomide was nearly abrogated following HLA blockade (Fig. 5E). Collectively, these data reinforce the possible association between the prognostic value of the CD27⁻:CD27⁺ ratio and the re-activation of clonal T cell expansions upon lenalidomide-based combination therapy.

## Discussion

Tumor recognition by T cells is essential for anticancer immunity and a deeper knowledge of clonal T cells is needed for informative immune profiling of patients. To our knowledge, we performed one of the largest studies of TILs in MM and its precursor states, which shed light into the evolving phenotype of large, expanded T cell clones during disease progression and identified an immune biomarker to predict survival in MM patients treated with lenalidomide-based combination therapy.

A major limitation to understand the origin and fate of T cells in tumor immunity is the lack of quantitative and qualitative information on the distribution of individual T cell clones[26]. Here, we showed that the extent of T cell clonality could be similar between healthy adults and patients with benign and malignant monoclonal gammopathies. This observation may be partially related to the fact that the BM acts as a reservoir of memory T cells, as well as a primary lymphoid organ[13], which could help explaining the presence of larger clones in healthy adults. In fact, the inability to discriminate BM-derived from blood-derived T cells is a limitation of the methods used in this study. However, the TCR repertoire in MGUS/SMM and MM patients was less diverse when compared to healthy adults, which would suggest that part of the intratumoral clones may recognize tumor antigens. In such a case, the long-lasting interaction between T cells and neoplastic plasma cells in the BM, would result in a more dysfunctional phenotype. The findings of altered

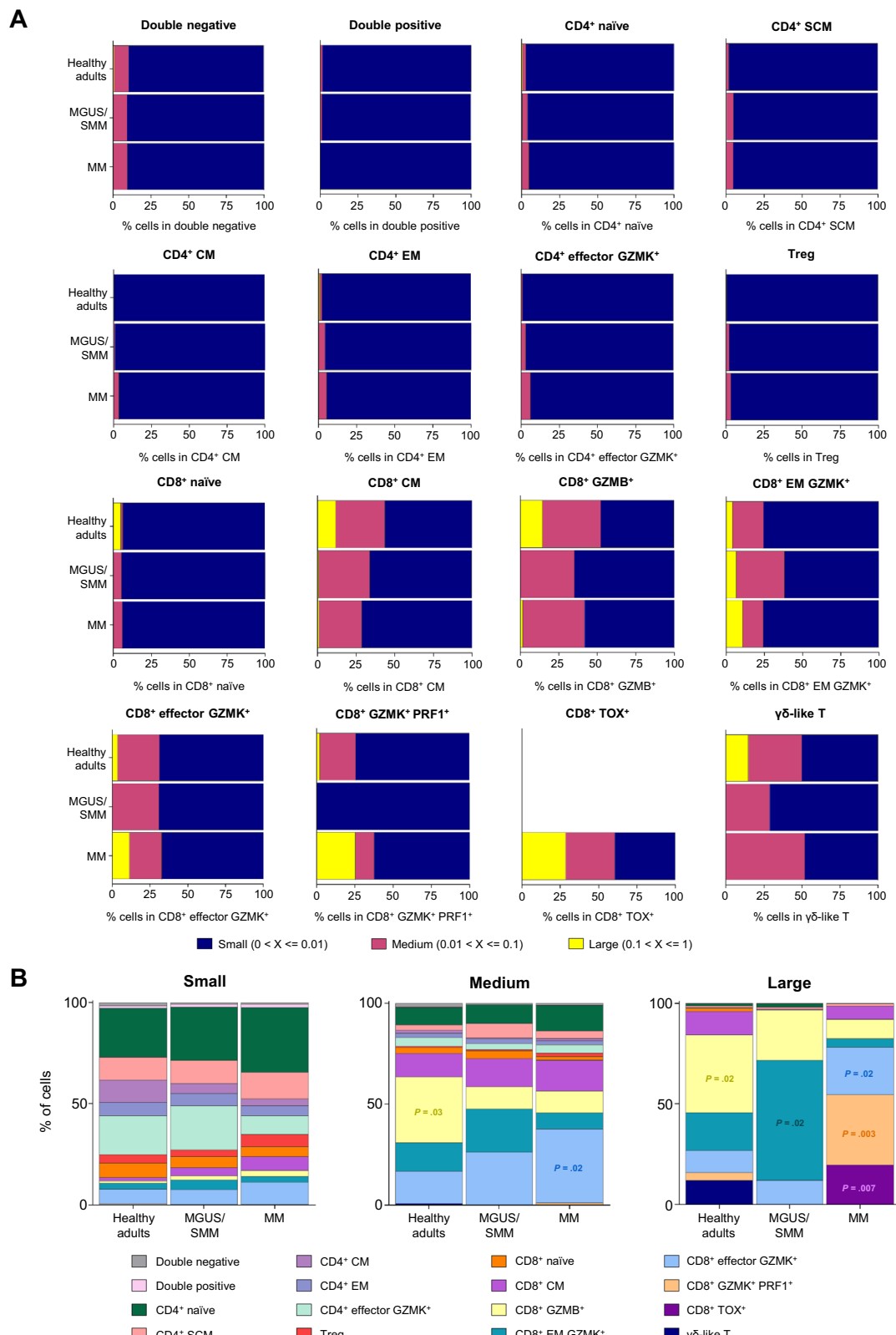

**Fig. 2 | Evolving phenotype of T cells during disease progression.**
**A** Transcriptional phenotype of small, medium and large T cell clones in bone marrow aspirates from healthy adults, MGUS/SMM and MM patients. T cells from the 16 clusters were categorized as small (range, 0–≤ 0.01), medium (range, 0.01–≤ 0.1) and large (range, 0.1–≤ 1) based on the percentage of each clonotype within total T cells and were classified according to their abundance in each clone group. Source data are provided as a Source Data file. **B** Distribution of the 16 T cell clusters among small, medium and large T cell clones in bone marrow aspirates from four healthy adults, eight MGUS/SMM and ten MM patients. *P* values were calculated using the Kruskal–Wallis test. Source data are provided as a Source Data file.

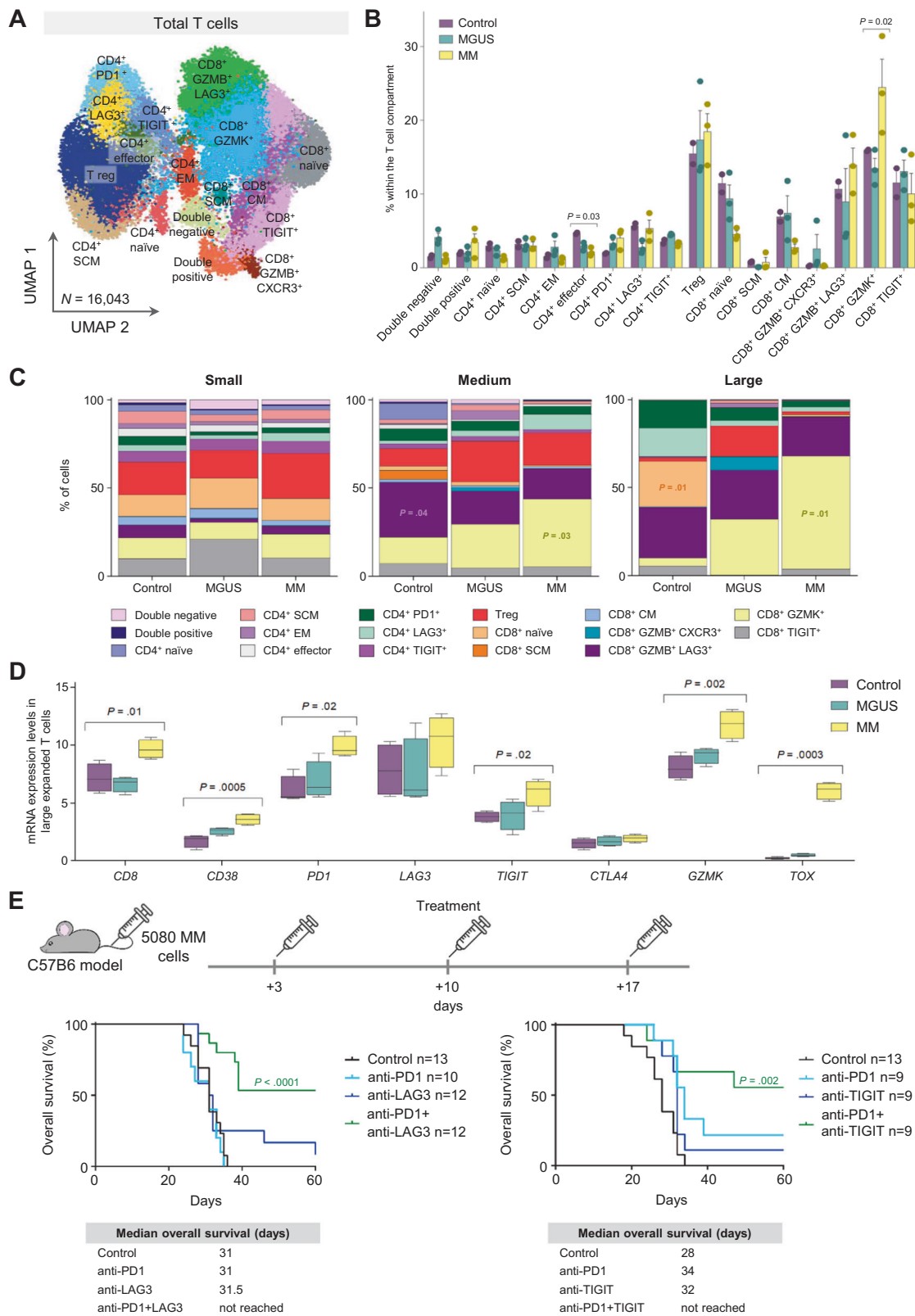

cluster distribution within large T cell clones from healthy adults vs. MGUS/SMM or MM patients, together with a higher proportion of cytotoxic over helper subsets, and the increased expression of consensus markers of T cell exhaustion in large clones from MM patients, support this hypothesis. As such, this study provides evidence about the mechanisms of tumor immune evasion during the progression of monoclonal gammopathies.

TILs include bystander and tumor-specific T cells with partially overlapping phenotypes[7,8]. Although selected markers have been utilized to exclude T cells recognizing a wide range of epitopes unrelated to cancer (e.g., PD-1, CD39, CD103), these may generate false-negative selection in tumors with limited data about their phenotype. Here, we provided a detailed list of 656 genes, including 35 coding for cell surface proteins, differentially expressed between small, medium and

**Fig. 3 | ICB combination therapy tailored to the phenotype of large T cell clones. A** Uniform manifold approximation and projection (UMAP) of 17 T cell clusters identified with single-cell RNA sequencing, in bone marrow aspirates from two control mice, three mice with monoclonal gammopathy of undetermined significance (MGUS) and three mice with active multiple myeloma (MM). **B** Relative distribution of the 17 clusters within the T cell compartment of control ($n = 2$), MGUS ($n = 3$) and MM ($n = 3$) mice. Error bars represent mean ± standard error mean (SEM). $P$ values were calculated using the Kruskal−Wallis test, *$p = 0.03$ and 0.02. Source data are provided as a Source Data file. **C** Distribution of the 17 T cell clusters among small, medium and large clones in bone marrow aspirates from control, MGUS and MM mice. $P$ values were calculated using the Kruskal−Wallis test. Source data are provided as a Source Data file. **D** mRNA expression levels of *CD8, CD38, PD1, LAG3, TIGIT, CTLA4, GZMK* and *TOX* in T cells with large T cell clones

in control ($n = 2$), MGUS ($n = 3$) and MM ($n = 3$) mice. Center and error bars represent median ± minimum and maximum. $P$ values were calculated using the Kruskal−Wallis test. Source data are provided as a Source Data file. **E** A total of $10 \times 10^6$ cells from the MM5080 cell line were intravenously injected into 8-week-old C57BL/6 mice. This cell line was established from bone marrow MM cells from a P53-BIcγ1 mouse, which results from the addition of a heterozygous P53 deletion to BIcγ1 mice. Three days after cell injection, mice were randomly divided into experimental groups and received a weekly dose of anti-PD1 (200 μg; RMP1-14), anti-LAG3 (200 μg; C9B7W) or anti-TIGIT (200 μg; 1G9), as monotherapy or in combination for the three following weeks. Kaplan−Meier overall survival for each group of mice is shown at the bottom of the panel. $P$ values were calculated using log-rank test.

large T cell clones from patients with monoclonal gammopathies. As expected, large clones showed overexpression of *CD8* and markers related to antigen-dependent differentiation such as *CD53, CD74, CD99, CD247* and *CXCR3*, as well as markers of T cell exhaustion such as *LAG3, PD-1,* and *TIGIT*. These data confirms and extends previous observations in MM[27], and could help explaining the limited clinical benefits of anti−PD-1 blockade in recent phase 3 clinical trials[16,17]. In particular, if large and possible tumor-reactive T cell clones express multiple immune checkpoints, anti−PD-1 alone may be insufficient to restore their function.

Whether ICB combination therapy is needed to reactivate and expand MM-specific T cells, similarly to solid tumors[28–31], remains unexplored in humans. Thus, we leveraged recently developed experimental models that recapitulate MM pathogenesis including mechanisms of immune evasion[20], to investigate the effect of ICB combination therapy in this tumor. As in humans, large T cell clones in mice exhibited potentially dysfunctional phenotypes involving the co-expression of various immune checkpoints at the MM stages. Interestingly, we observed prolonged survival with anti-PD1 plus anti-LAG3 or anti-TIGIT combination therapy. These results build upon previous pre-clinical evidence of an immune-inhibitory role of LAG3 and TIGIT in MM, and a possible benefit when blocking these immune checkpoints[32–34].

Lenalidomide is a backbone in the treatment of newly-diagnosed MM patients[35,36]. IMIDs bind to the E3 ligase substrate-recognition adapter protein cereblon and, therefore, the protein is essential for the therapeutic effect of these drugs. Surprisingly though, more than two thirds of patients do not show point mutations, copy losses/structural variations or specific variant transcripts of cereblon by the time they become refractory to IMIDs[37]. Thus, determinants of response and resistance remain largely unknown and there are no routine bio-markers to predict clinical outcomes prior lenalidomide treatment[38–41]. Our results suggest that the efficacy of IMIDs could depend on the extent of clonal T cell expansions. Namely, we showed that the ratio between CD27$^-$ and CD27$^+$ T cells in the TME predicted progression-free survival in two independent series of patients, representative of both transplant-eligible and ineligible MM. The lack of impact of the CD27$^-$:CD27$^+$ T cell ratio in the OS of transplant-ineligible patients could be related to the different treatment protocols. In the GEM-CLARIDEX clinical trial, patients received lenalidomide-based combi-nations until disease progression; hence, these were considered as IMID-refractory at relapse which could have limited the options of salvage regimens. By contrast, transplant-eligible patients enrolled in the GEM2012MENOS65 clinical trial received fixed-duration therapy and might have been eligible to IMID-containing salvage regimens at relapse.

One limitation of this study is the lack of paired MFC and scRNA/TCR-seq (or bulk TCR-seq) data in a large series of MM patients to compare the prognostic value of the CD27$^-$:CD27$^+$ T cell ratio with other metrics of T cell clonality. Thus far, scRNA/TCR-seq data has been generated in small cohorts[11,12], which probably reflects the

economical challenge of performing these methods in large series. Indeed, this is what elicited us to investigate phenotypic hallmarks of large intratumoral T cell clones to rely on more commonly available MFC data. Of note, CD27 is a well-known marker of T cell activation and antigen-dependent differentiation, and there might be better surrogates of clonal T cell expansions (Fig. 4A and Supplementary Dataset 5). We specifically selected CD27 for subsequent analysis because of its inclusion in the MFC panels used for the screening of newly-diagnosed MM patients.

A strength of the observed relationship between the CD27$^-$:CD27$^+$ T cell ratio and outcome, is that it is agnostic to the clone target. This is important because the clonal T cell repertoire will vary markedly within and between individuals and thus, a generalized surrogate biomarker is required for translation to patient care. Future studies, namely those including more recent standards of care such as anti-CD38 monoclonal antibodies, will determine the extent to which the CD27$^-$:CD27$^+$ T cell ratio is a useful maker of tumor specificity and response to lenalidomide-based combinations in MM. As noted above, CD27 is commonly used to screen for plasma cell clonality in patients with monoclonal gammopathies[21,22], and MFC is considered as an obligatory diagnostics test in MM[35]. Thus, there is potential for the CD27 ratio to become a biomarker of patients with different T cell composition who may display distinct clinical behavior upon treat-ment with lenalidomide. Such hypothesis should be confirmed in additional series, particularly those treated with more recent immunotherapies.

As human cancers arise in an immunocompetent host, tumor development is shaped by immunoediting and malignant cells develop the capacity to escape tumor antigen responses[42]. Indeed, the classical paradigm of host-tumor interaction−i.e., elimination, equilibrium and escape −, is reflected in the clinical behavior of MM which progresses from MGUS and SMM[13]. Our results suggest that large T cell clones in patients with precursor conditions appear to be less dysfunctional than those from cases with active disease. If confirmed in larger series, this finding could help to better understand the effects of lenalidomide (alone or in combination with dexamethasone) in modulating T cell phenotypes[43] and in prolonging time-to progression of high risk SMM patients treated in two phase 3 clinical trials[14,15]. Therefore, this study illustrates the power of combining transcriptomic with TCR analysis to shed light on the functional impairment of T cell mediated immunity in the TME, and how the systematic interrogation of TILs is key to future development of immunotherapy, and the prediction of clinical responses in cancer.

## Methods
### Patients and treatment
A total of 565 BM samples from healthy adults ($n = 4$) or patients with MGUS ($n = 4$), SMM ($n = 4$) or newly-diagnosed MM ($n = 553$) were analyzed (median ages of 62, 68, 47, and 66 years, respectively). Only samples with >90% viability (according to the percentage of debris observed by flow cytometry) were used for analysis. Of the 553 MM

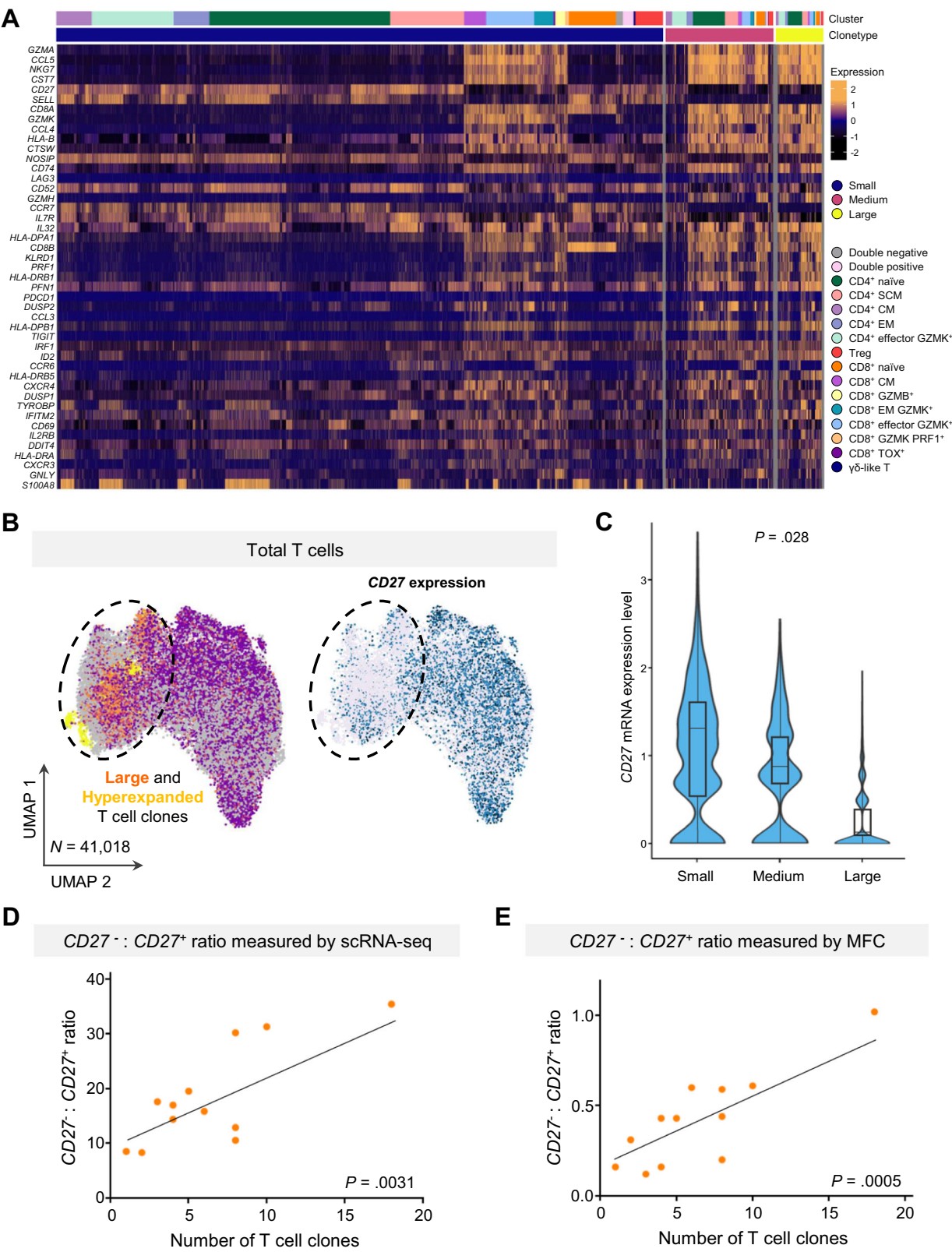

patients, 272 were enrolled in the PETHEMA/GEM2012MENOS65 clinical trial and 271 in the PETHEMA/GEMCLARIDEX clinical trial (registered at http://www.clinicaltrials.gov as #NCT01916252 and #NCT02575144, respectively). These two cohorts were selected to determine the prognostic value of the CD27 ratio measured in intratumoral T cells. Briefly, in the PETHEMA/GEM2012MENOS65 clinical trial, patients received six induction cycles of bortezomib,

lenalidomide and dexamethasone, followed by autologous stem-cell transplantation conditioned with Bu-Mel or Mel-200 high-dose therapy, and received two consolidation cycles of bortezomib, lenalidomide and dexamethasone. In the PETHEMA/GEMCLARIDEX clinical trial, patients received either clarithromycin, lenalidomide and low-dose dexamethasone or lenalidomide and low-dose dexamethasone. This study was approved by the Institutional Review Board of the

**Fig. 4 | T cell markers of clonality in MM. A** Heatmap of the most differentially expressed genes between T cells with small, medium and large T cell clones from four healthy adults, eight patients with benign monoclonal gammopathy of undetermined significance (MGUS) or smoldering multiple myeloma (SMM), and ten patients with active, newly-diagnosed multiple myeloma (MM). A log-transformed fold-change was used to measure gene expression. mRNA expression of *CD27* in bone marrow T cells with small, medium and large T cell clones shown in a (**B**) uniform manifold approximation and projection (UMAP) and (**C**) violin plot. Center and error bars of the boxplots represent median ± minimum and maximum.

*P* values were calculated using the Kruskal–Wallis test. **D** Association between the ratio of *CD27* negative and positive (*CD27*⁻ : *CD27*⁺) T cells with the number of T cell clones analyzed by single-cell RNA and TCR sequencing (scRNA/TCR-seq). *P* values were calculated using the two-sided Pearson's correlation test. Source data are provided as a Source Data file. **E** Association between the *CD27*⁻ : *CD27*⁺ ratio with the number of T cell clones analyzed by multidimensional flow cytometry (MFC). *P* values were calculated using the two-sided Pearson's correlation test. Source data are provided as a Source Data file.

University of Navarra (2017.022) and was conducted according to the principles of the Declaration of Helsinki. Informed consent was obtained from all participants.

## Combined single-cell RNA and TCR sequencing (scRNA/TCR-seq)

scRNA-seq + scTCR-seq were performed in 22 BM aspirates from 4 healthy adults, 4 MGUS, 4 SMM, and 10 MM patients, as well as 8 BM aspirates from 2 control (Y$_{cy1}$), 3 MGUS (BI$_{cy1}$) and 3 MM (BI$_{cy1}$) bearing mice. In order to discriminate BM samples with peripheral blood contamination, we implemented a quality control check, which is to analyze the presence of BM specific cell types using flow cytometry: B cell precursors, mast cells and nucleated red blood cells. If these are absent, we conclude that the level of hemodilution is high and the sample cannot be considered representative of a BM aspirate. We compared the data from our samples with a large reference dataset[19] and only used those in which the percentages of these cell populations were inside the reported ranges (Table S2). Cells were FACS sorted (a mix of $0.8 \times 10^5$ CD3⁺ T cells + $0.8 \times 10^5$ CD56⁺ NK cells + $0.25 \times 10^5$ CD300e⁺ monocytes + $0.15 \times 10^5$ CD19⁺ B cells) in 100 μL of PBS + 0.05% BSA. Samples with at least 90% viability were processed using the 10X Genomics (CA, USA) scRNA/TCRseq kit, following the manufacturer's instructions (Chromium Next GEM Single Cell V(D)J v1.1 protocol rev F for human samples and Chromium Next GEM Single Cell 5′ v2 Dual Index protocol rev B for mice samples). Quality control was performed with Qubit Fluorometric Quantification (ThermoFisher Scientific, MA, USA) using the double-stranded DNA high-sensitivity assay kit, and with TapeStation (Agilent, Santa Clara, CA) using high-sensitivity screentapes. Libraries were sequenced on a NextSeq 550 (Illumina, San Diego, CA) with the sequencing depth and run parameters indicated by 10X Genomics instructions. scRNAseq data is available at the NCBI GEO under accession GSE205393.

scRNA-seq and scTCR-seq data from humans and mice were analyzed separately. Sample demultiplexing, alignment to the hg38 human reference genome (or the respective mice genome) and single-cell gene count was performed using the Cell Ranger Single-Cell Software Suite v.6.0 (https://www.10xgenomics.com/). Expression matrixes were analyzed with the R package *seurat* 4.0 (https://satijalab.org/seurat/) and cells were filtered according to <10% mitochondrial expression and at least 200 (but less than 2500) mRNA counts per cell. Once scaled and normalized, a genelist including the most variable genes was obtained by the FindVariableFeatures function. After removing genes belonging to the immunoglobulin families (which could work as a confounding factor during clustering), the genelist was used to derive the principal component analysis (PCA) vectors for each sample. The first 100 PCA were used to align samples (batch removal) using the R package *harmony* v.0.1.1[44]. The new harmonized coordinates were used to develop UMAPs (dimensionality reduction). The shared nearest neighbor (SNN) algorithm based on 50 batch-corrected dimensions was used for clustering the cells into homogeneous groups that were manually identified according to the expression of canonical genes (see Supplemental Information) obtained from curated gene-sets. A sequential subclustering strategy (which consists basically in repeating the same steps as before on a specific subpopulation) to focus on clonotypic T cells was performed. Annotation of T cell

clusters was performed taking into account the expression of genes reported in Fig. S1.

T cell clones defined by their unique CDR3 of both α and β chains were obtained from Cell Ranger v.6.0 and using the *scRepertoire* v.1.10.1 R package. This information was added to the Seurat object to analyze the transcriptome of these cells. *scRepertoire* R package was also used to assess clonotype distribution as well as to investigate clonal "diversity", characterized by clones frequency, repertoire richness and convergence[45,46]. To estimate clonal expansions and richness we used Shannon, Inverse Simpson, Chao and ACE indices[47–50] which have been developed to deal with under sampling (i.e., "unseen species"), and could therefore mitigate the fact that for technical reasons, only a fraction of repertoires is sequenced and analyzed[45].

## Multidimensional flow cytometry (MFC)

The phenotype of T cells in BM samples from newly-diagnosed MM (*n* = 553) was analyzed using MFC. Samples were stained following the EuroFlow lyse, wash and stain standard sample preparation protocol, adjusted to 10⁶ nucleated cells. EDTA-anticoagulated BM aspirates were stained with the following combination of the monoclonal antibodies: CD138-BV421, CD27-BV510, CD38-FITC, CD56-PE, CD45-PerCPCy5.5, CD19-PECy7, CD117-APC, and CD81-APCH7 (Table S8).

## Computational flow cytometry

FCS files from 553 BM aspirates from newly diagnosed MM patients were analyzed using the semi-automated algorithm named "FlowCT" v.0.0.9[51], which is based on the analysis of multiple files by automated cell clustering. Briefly, FCS files were merged, underwent quality control, were normalized through batch removal steps and clustered using FlowSOM. After the computational clustering, the Infinicyt software v.2.0 (Cytognos SL, Salamanca, Spain) was used for the identification of each cluster. Statistical analysis was then performed based on the output of the software.

## RNA sequencing (RNA-seq)

Normal plasma cells from healthy adults (*n* = 25) and tumor cells from MGUS (*n* = 12) and newly-diagnosed MM patients (*n* = 216) were isolated from total BM aspirates, in a FACSAriaII and according to patient-specific aberrant phenotypes. Bulk RNA-seq was performed using a protocol adapted from single-cell massively parallel single-cell RNA-sequencing[52], which enabled preparing libraries with as few as 20.000 cells as starting material. Briefly, we barcoded RNA from each sample in a retrotranscription (RT) reaction with AffinityScript Multiple Temperature Reverse Transcriptase (Agilent) and different RT primers. After qPCR, cDNA with similar Ct values were pooled together. cDNA was purified with SPRIselect 1.2X (Beckman Coulter -BC-, Brea, CA) and amplified using the T7 promotor as template previously introduced in the RT reaction. T7 polymerase (NEB) was added for 16 h at 37 °C. RNA molecules were fragmented with 2 μL of 10X Zn²⁺ fragmentation buffer (Ambion™, ThermoFisher Scientific) for 1 min at 70 °C and purified with SPRIselect 2X. Afterwards, a ssRNA adapter (Illumina) was ligated to the 3′-end of the RNA fragments in presence of DMSO, 100 mM ATP, 50% PEG and T4 RNA ligase I (NEB) for 2 h at 22 °C. A second RT reaction was performed with AffinityScript Multiple Temperature Reverse Transcriptase and resulting cDNA was purified with SPRIselect

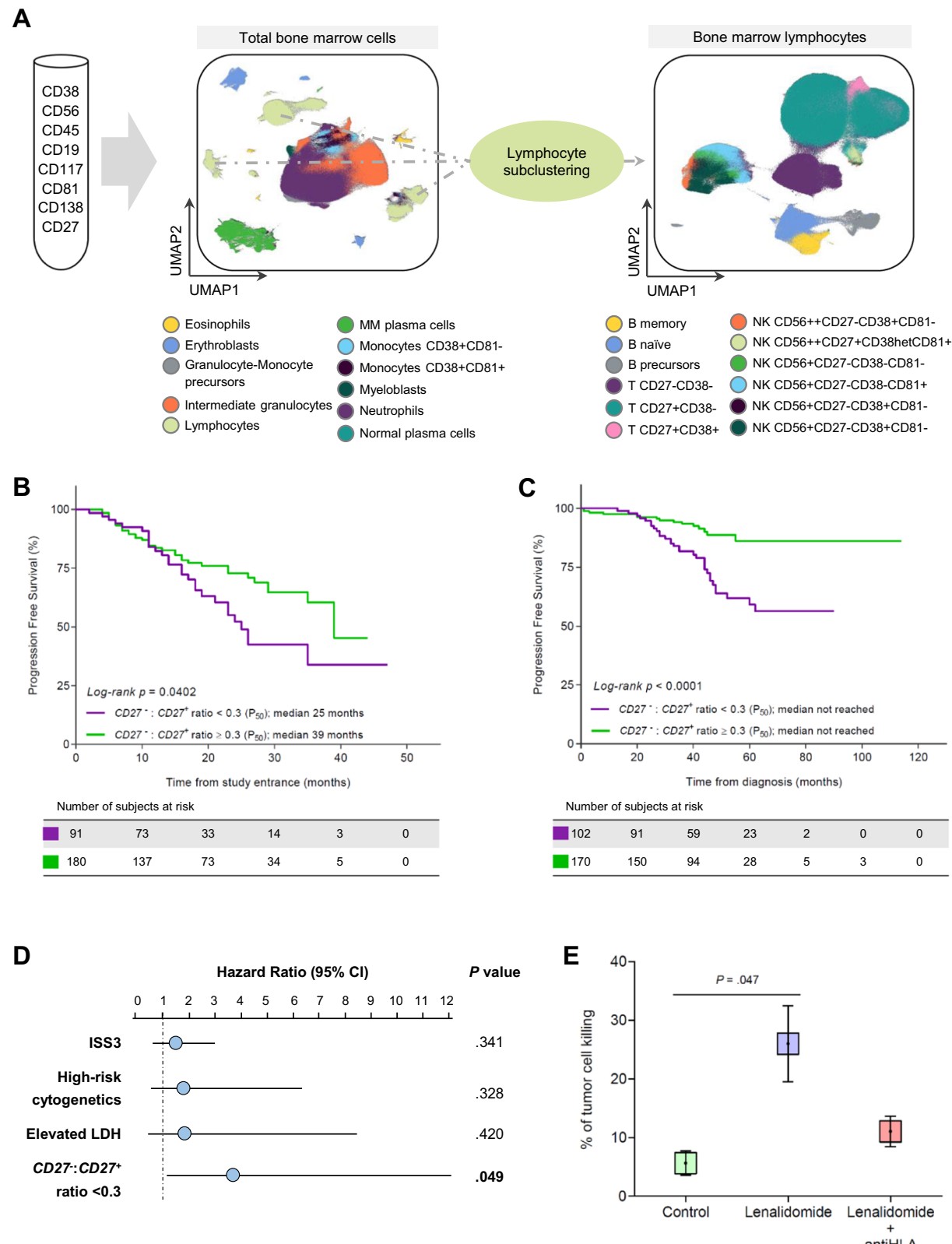

1.5X. Finally, cDNA was amplified with 12.5 μL Kappa Hifi ready mix + 1 μL 25 μM primer mix per sample and purified with SPRIselect 0.7X. Qubit, TapeStation and qPCR analysis were done as quality controls and 4 nM of the final library were sequenced in a NextSeq 550 (Illumina). Differential gene expression across all pairwise comparisons between groups was analyzed with *Deseq2* v.1.40.2 R package followed by k-means clustering of genes in R.

## Whole-exome sequencing (WES)

We performed WES in tumor cells from BM aspirates and peripheral blood T cells of MM patients (*n* = 23). We sorted approximately 20,000 tumor cells in 100 μL of Lysis/Binding Buffer (ThermoFisher Scientific, MA, USA), using the same strategy described above for bulk RNA-seq. The quality of genomic DNA extracted from tumor and peripheral blood T cells was evaluated by Agilent 4200 Tape Station using

**Fig. 5 | T cell markers of progression in MM. A** Uniform manifold approximation and projection (UMAP) of bone marrow cells from newly-diagnosed multiple myeloma (MM) patients. All samples were stained with the same eight-color monoclonal antibody combination described in the panel, and processed using standardized protocols. Computational flow cytometry was used to cluster bone marrow cells and to subcluster lymphocytes. A total of 22 clusters and subclusters were identified, including *CD27* negative and positive T cells. **B** Progression-free survival of 271 transplant-ineligible MM patients enrolled in the PETHEMA/GEM-CLARIDEX clinical trial, stratified according to values of *CD27⁻* : *CD27⁺* ratio in T cells below vs equal or greater than the median value observed in the entire MM series (0.3). **C** Progression-free survival of 272 transplant-eligible MM patients enrolled in the PETHEMA/GEM2012MENOS65 clinical trial, stratified according to values of the *CD27⁻* : *CD27⁺* ratio in T cells below vs equal or greater than the median value

observed in the entire MM series (0.3). **D** Multivariate analysis of progression-free survival (PFS) considering established risk factors at diagnosis (i.e., International Staging System [ISS] 3, high-risk cytogenetics defined by the presence of t(4;14), t(14;16) and/or del(17p), and elevated lactate dehydrogenase [LDH] levels) and the *CD27⁻* : *CD27⁺* ratio in T cells from MM (*n* = 543) patients. Blue dots represent the hazard ratio and bars represent the 95% confidence interval. Hazard ratio and 95% CI were determined using the regression coefficient of the Cox model. **E** Boxplots representing the percentage of tumor cell killing after culture in a 3D organoid model of the bone marrow of MM patients (*n* = 3) treated or not with 1 μM lenalidomide, with or without an anti-HLA antibody. Center and error bars represent mean ± SEM. *P* values were calculated using a two-sided Student's *t* test. Source data are provided as a Source Data file.

Genomic DNA ScreenTape system (Agilent, USA), and DNA concentration quantified by Qubit System (Invitrogen, USA). Genomic DNA was captured for each sample in a 10X Chromium instrument using the Chromium Genome Reagent Kit V2 for Exome Assays (10X Genomics, USA). DNA was then fragmented to an average size of 225 bp using a Covaris S220 ultrasonicator (Covaris, USA) and subjected to DNA library construction using Chromium Genome Reagent Kit V2 for Exome Assays (10X Genomics, USA). Target enrichment was performed with SureSelectXT Human All Exon V6 Capture Library (Agilent Technologies, USA) and sequence targets were captured and amplified in accordance with manufacturer's recommendations. Enriched libraries were used for 150 base sequencing in a NovaSeq 6000 (Illumina, USA) following manufacturer's instructions. Raw FASTQ files were processed using LongRanger (v2.2.2, 10xGenomics) with default parameters. Variants were annotated using the bioinformatics software HD Genome One (DREAMgenics, Oviedo, Spain), using several databases containing functional (Ensembl, CCDS, RefSeq, Pfam), populational (dbSNP, 1000 Genomes, ESP6500, ExAC) and cancer-related (COSMIC−Release 87, ICGC−Release 27) information. In addition, 9 scores from algorithms for prediction of the impact caused by non-synonymous variants on the structure and function of the protein were used (SIFT, PROVEAN, Mutation Assessor, Mutation Taster, LRT, MetaLR, MetaSVM, FATHMM, and FATHMM-MKL), and 1 score (GERP + +) for evolutionary conservation of the affected nucleotide. Indel realignment was performed to correct underestimated allele frequencies. Variants with a population allele frequency higher than 0.01 were excluded. Variants detected in germline DNA (i.e., T cells) were excluded. Only mutations with a coverage higher than 6 in all samples from a patient were selected. Only variants detected in a sample with a variant frequency >=0.15, with a mutated allele count >=4 and droplet count >=4 were selected. Class A HLA haplotypes were identified using optiType (v1.3.3) genotyping algorithm[53].

## Mouse model of MM for scRNAseq + scTCRseq analysis

B6(Cg)-*Gt(ROSA)26Sor*^tm4(Ikbkb)Rsky^/J mice (stock #008242) with constitutively active NF-κB signaling by IKK2 expression and a green fluorescent protein (GFP) reporter[54], B6.Cg-Tg(BCL2)²²Wehi/J mice (stock #002319) with BCL2 expression[55], and B6.129P2-*Trp53*^tm1Brn^/J mice (stock #008462) with p53 deletion[56] were obtained from The Jackson Laboratory (Bar Harbor, ME, USA). Transgenic activation in germinal center B lymphocytes was obtained by crossing mice with the cre-recombinase mouse line cγ1-cre (B6.129P2(Cg)-*Ighg1*^tm1(cre)^ ^Cgn^/J, stock #010611) obtained from the Jackson laboratory[57]. Strains were intercrossed by conventional breeding to obtain the corresponding compound mice with heterozygous alleles, termed BIcγ1, as this carries BCL2 and IKK2 expression by the cγ1-cre allele, and P53-BIcγ1, which also carries P53 deletion. BIcγ1 mice (*n* = 20) and P53-BIcγ1 mice (*n* = 20) consistently developed human-like MM, with a median OS of 296 and 258 days, respectively. As controls, cγ1-cre mice crossed to B6.129X1-Gt(ROSA)*26Sor*^tm1(EYFP)Cos^/J mice (stock #006148, The Jackson Laboratory), which carry a yellow fluorescent

protein (YFP) reporter, were also generated (*n* = 20)[58]. All mice were maintained in a hybrid C57BL6/129 Sv genetic background under specific pathogen-free conditions in the animal facilities of the Center for Applied Medical Research CIMA at the University of Navarra. To induce the formation of GFP⁺ transgenic plasma cells, animals were subjected to T cell-mediated immunization with sheep red blood cells (SRBCs) intraperitoneally administered at 8 weeks of age, and then repeated every 21 days for 4 months. After immunization, mice were monitored twice a week for tumor development or end-point criteria such as reduction of mobility, labored respiration, or bristly hair. Mice were euthanized by cervical dislocation when signs of disease appeared, being then characterized. To this end, flow cytometry was applied to cell suspensions from bone marrow (flushed from femurs with DPBS) with the following murine monoclonal antibodies to detect tumor and immune cell subpopulations: CD138-PE, B220-APC, CD19-APC-Cy7, IgM-BV421, CD3-PE-Cy7, CD4-APC, CD8-BV510, NK1.1-BV421, FoxP3-PE, CD25-BV510, PD1-BV421, TIGIT-PE, LAG3-APC-Cy7, CD11b-BV510 and Gr1_PE-Cy7 (Table S8). Data acquisition was performed in a FACS CantoII flow cytometer (BD Biosciences) and analyzed using FlowJo™ V10.7.1 software. In addition, serum protein electrophoresis of blood samples was used to measure the gamma-globulin (γ) fraction in a semi-automated Hydrasys 2 device, along with an isotyping multiplex assay to quantify Ig isotypes in serum using the MILLIPLEX® Mouse Immunoglobulin Isotyping kit on the Luminex® xMAP® platform. Tumor clonality was determined by genomic amplification of *IgHV* gene sequences by PCR in DNA isolated from GFP + -sorted MM cells in diseased mice, using specific VHA, VHE, and VHB forward primers and a reverse primer for JH4. Survival rates of mice were estimated using Kaplan−Meier overall survival curves. Mice of both sexes were used in the study and were kept under specific-pathogen-free conditions in the animal facilities of the Center for Applied Medical Research (CIMA) at the University of Navarra. Animal experimentation was approved by the Ethical Committee of Animal Experimentation of the University of Navarra and by the Health Department of the Navarra Government.

## In vivo pre-clinical trial in a syngeneic MM model using immune checkpoint inhibitors

The MM-derived MM5080 cell line was established from an original MM developed in P53-BIcγ1 mice. Establishment of syngeneic transplants was performed by injecting 10 × 10⁶ MM5080 cells in DPBS into the tail veins of 8- to 10-week-old immunocompetent C57BL/6JOlaHsd mice (strain code: 057, Envigo). Three days upon injection of MM cells, animals of both sexes were randomly divided into experimental groups, which received anti-PD1 (200 μg; RMP1-14), anti-TIGIT (200 μg; 1G9), or anti-LAG3 (200 μg; C9B7W) administered intraperitoneally alone or in combination for the following 3 weeks, while control mice received vehicle. Therapy responses were estimated by Kaplan−Meier survival curves, which were compared using the log-rank test using v7 GraphPad Prism software.

## 3D cultures

A 3D organoid was developed to test the effect of lenalidomide (1 μM), alone or in combination with 10 μg/mL of an anti-HLA I (BioXCell, Lebanon, NH), on tumor plasma cell killing from bone marrow aspirates of MM patients ($n = 3$). Cells were lysed with 1X BulkLysis buffer (Cytognos) and $5 \times 10^6$ cells were embedded in 60 μL of Matrigel Matrix (Corning) and fibronectin (ratio matrigel:fibronectin 2:1). This mix was seeded per well in a 24-well plate (Cellstar®) and left 40 min in the incubator, so that the matrigel may solidify Afterwards, we added 1 mL of RPMI1640 medium (10% FBS, 1% L-Glu, 1% Penicillin-Streptomycin) supplemented with 10% of plasma from the same BM sample, IL-6 100 nM and IGF1 100 nM per well. Organoids were maintained in culture for 5 days at 37 °C. Finally, organoids were desegregated with Cell Recovery Solution (Corning) and labeled with CD138-BV421, CD3-BV510, CD38-FITC, CD4-PE, CD45-PerCP-Cy5.5, CD19-PE-Cy7, AnnexinV-APC and CD8-APCH7 (Table S8). Data acquisition was performed in a FACSCantoII flow cytometer using the FACSDiva software and data analysis was performed using the Infinicyt software.

## Statistical analysis

Survival probabilities were estimated by using the Kaplan–Meier method and compared with the use of a two-sided stratified log-rank test. The effect of CD27 ratio on the risk of progression-free survival (PFS) and overall survival (OS) (hazard ratio [HR]), with its two-sided 95% confidence interval (CI), were estimated with a logistic Cox regression model. PFS was defined from the time of MFC assessment at diagnosis until disease progression or death from any cause. OS was defined from the time of MFC assessment at diagnosis until death. A multivariate Cox proportional hazard model was developed to explore the independent effect on PFS of prognostic factors defining the revised International Staging System (R-ISS). Patients were stratified into groups according to the median value of the CD27 ratio or the abundance of T cell subsets in the whole population. The Kruskal–Wallis test was used to estimate the statistical significance observed between groups in the comparison between T cells ranging from healthy adults to newly-diagnosed MM patients. Student's $T$ test was used to evaluate differences between groups in the 3D culture experiment. Statistical analyses were performed using the GraphPad Prism software (version 7, San Diego, CA), R version 4.0 and SPSS (version 25.0.0, IBM, Chicago, IL). $P < 0.05$ were considered as statistically significant.

## Reporting summary

Further information on research design is available in the Nature Portfolio Reporting Summary linked to this article.

## Data availability

scRNAseq read data are submitted at NCBI GEO under accession GSE205393. hg38 human and mm10 mice reference genome were obtained from https://support.10xgenomics.com/single-cell-gene-expression/software/release-notes/build. Source data are provided with this paper.

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

## Acknowledgements

This work was supported by grants from the Instituto de Salud Carlos III/Subdireccion General de Investigacion Sanitaria and co-financed by FEDER funds (CB16/12/00233, CB16/12/00369, PI17/01243, PI19/00818 and PI20/00048), and together with Fundación Científica de la Asociación Española Contra el Cáncer (FCAECC) for iMMunocell Transcan-2 (AC17/00101), FCAECC Predoctoral Grant Junta Provincial Navarra, the Cancer Research UK (C355/A26819), FCAECC and Italian Association for Cancer Research (AIRC) under the Accelerator Award Program (EDITOR), 2017 Multiple Myeloma Research Foundation Immunotherapy Networks of Excellence, Black Swan Research Initiative of the International Myeloma Foundation, European Hematology Association nonclinical advanced research grant (3680644), European Research Council 2015 Starting Grant (MYELOMANEXT grant 680200), the Cancer Research Innovation in Science Cancer Foundation (PR_EX_2020-02), the Leukemia Lymphoma Society, unrestricted grants from Bristol-Myers Squibb/Celgene and Takeda, Roche imCORE program (NAV-4 and NAV-15), Fondazione Regionale per la Ricerca Biomedica (Regione Lombardia) (Project ID 065 JTC 2016), ERA-NET TRANSCAN-2, and by My First AIRC Grant 2020 (n. 24534, 2021/2026), and by the Riney Family Multiple Myeloma Research Program Fund.

## Author contributions

C.B., C.P., and B.P. conceived of the idea and designed the study protocol; C.P. performed scRNA-seq and scTCR-seq experiments; C.P. and S.S. performed next-generation sequencing; S.R. performed exome sequencing experiments; C.B., C.P., and I.G. analyzed sequencing data; C.B., C.P., N.P., M.-T.C., R.T., and A.Z. analyzed flow cytometric data; A.L. performed cell sorting; M.L. performed in vivo experiments; C.P. and L.B. performed in vitro experiments; D.M.P., M.-N., and D.B. developed the machine learning model; M.G., S.S., A.Oriol, M.E.G.-G., A.S., F.d.A., R.R.T., J.-M.M., M.G., M.T.H., J. Bargay, L.P., A.P.-M., H.G., H.A.-L., A.R., A. Orfao, J.M.-L., J.-J.L., L.R., J. Blade, M.-V.M., J.-A.M.C., and J.F.S.-M. provided study material and/or patients and/or support; C.B. and C.P. performed statistical analyses; C.B., C.P., and B.P. wrote the manuscript; and all authors reviewed and approved the manuscript.

## Competing interests

C.B. has served as a member on advisory boards for Amgen, Janssen, Pfizer, Takeda, and Oncopeptides; A.R. has received honoraria from Amgen, Celgene, and Janssen and a research grant from AstraZeneca and the Associazione Italiana per la Ricerca sul Cancro (AIRC): AIRC-IG-24689. H.G. has received speakers bureau honoraria from Academy2, KG, Agentur Hogg Robinson Germany, Amgen, ArtTempi, Beupdated Helbig Consulting and Research AG Schweiz, Bristol Myers Squibb, Celgene, Chop, Chugai, Congress Culture Concept Dr. S. Stocker München, Connectmedia Warschau/Polen, Dr. Hubmann Tumorzentrum München, FomF, GlaxoSmithKline, GWT Forschung und Innovation Dresden, Institut für Versorgungsforschung in der Onkologie GbR, Janssen, Kompetenznetz Maligne Lymphome, MedConcept, Medical Communication, Münchner Leukämie Labor Prof. Haferlach, New Concept Oncology, Novartis, Omnia Med Deutschland, Onko Internetportal DKG-web, Sanofi, STIL Forschungs, and Veranstaltungskonzept Gesundheit Mechernich, has served as a member on advisory boards for Adaptive Biotechnology, Amgen, Bristol Myers Squibb, Celgene, Janssen, Sanofi, and Takeda, and has received research grants and/or materials such as investigational medicinal products from Amgen, Bristol Myers Squibb, Celgene, Chugai, Dietmar-Hopp-Foundation, Janssen, John Hopkins University, and Sanofi. A. Oriol participated in advisory boards for Amgen, Celgene and Janssen. M.-V.M. has received honoraria for lectures from or participated in advisory boards for Janssen, Celgene, Amgen, Takeda, AbbVie, Adaptive, GSK, Pharmamar, EDO, and Oncopeptides. L.R. reports honoraria from Janssen, Celgene, Amgen, and Takeda. J.B. reports honoraria for lectures from Janssen, Amgen, Celgene, Takeda, and Oncopeptides. J.-J.L. reports honoraria from and membership on boards of directors or advisory committees with Takeda, Amgen, Celgene, and Janssen. J.F.S.-M. reports consultancy for Bristol-Myers Squibb, Celgene, Novartis, Takeda, Amgen, MSD, Janssen, and Sanofi and membership on a board of directors or advisory committee with Takeda. J.A.M.-C. has received research grants from Roche, Bristol-Myers Squibb-Celgene, and Janssen. B.P. reports honoraria for lectures from and membership on advisory boards with Adaptive, Amgen, Becton Dickinson, Bristol-Myers Squibb-Celgene, Janssen, Merck, Novartis, Roche, Sanofi and Takeda; unrestricted grants from Bristol-Myers Squibb-Celgene, EngMab, Roche, Sanofi, and Takeda; and consultancy for Bristol-Myers Squibb-Celgene, Janssen, Sanofi, and Takeda. The remaining authors declare no competing interests.

## Additional information

Cirino Botta [1,2,26] ✉, Cristina Perez [2,26], Marta Larrayoz [2], Noemi Puig[3], Maria-Teresa Cedena [4], Rosalinda Termini[2], Ibai Goicoechea [2], Sara Rodriguez [2], Aintzane Zabaleta[2], Aitziber Lopez[2], Sarai Sarvide[2], Laura Blanco[2], Daniele M. Papetti[5], Marco S. Nobile [6,7], Daniela Besozzi[5,7], Massimo Gentile[8], Pierpaolo Correale[9], Sergio Siragusa[1], Albert Oriol[10], Maria Esther González-Garcia[11], Anna Sureda[12], Felipe de Arriba[13], Rafael Rios Tamayo[14], Jose-Maria Moraleda [13], Mercedes Gironella[15], Miguel T. Hernandez[16], Joan Bargay[17], Luis Palomera[18], Albert Pérez-Montaña[19], Hartmut Goldschmidt [20], Hervé Avet-Loiseau[21], Aldo Roccaro [22], Alberto Orfao [23,24], Joaquin Martinez-Lopez[4], Laura Rosiñol [25], Juan-José Lahuerta [4], Joan Blade [25], Maria-Victoria Mateos [3], Jesús F. San-Miguel [2], Jose-Angel Martinez Climent [2], Bruno Paiva [2] ✉, the Programa Para el Estudio de la Terapéutica en Hemopatías Malignas/Grupo Español de Mieloma (PETHEMA/GEM) cooperative group* and the iMMunocell study group

[1]Department of Health Promotion, Mother and Child Care, Internal Medicine and Medical Specialties, University of Palermo, Palermo, Italy. [2]Clinica Universidad de Navarra, Centro de Investigacion Medica Aplicada (CIMA), CCUN, Instituto de Investigacion Sanitaria de Navarra (IDISNA), CIBER-ONC numbers CB16/12/00369, CB16/12/00489, Pamplona, Spain. [3]Hospital Universitario de Salamanca, Instituto de Investigacion Biomedica de Salamanca (IBSAL), Centro de Investigación del Cancer (IBMCC-USAL, CSIC), CIBER-ONC number CB16/12/00233, Salamanca, Spain. [4]Hospital Universitario 12 de Octubre, CIBER-ONC number CB16/12/00369, Madrid, Spain. [5]Department of Informatics, Systems and Communication, University of Milano-Bicocca, Milan, Italy. [6]Department of Environmental Sciences, Informatics and Statistics, Ca' Foscari University of Venice, Venice, Italy. [7]Bicocca Bioinformatics, Biostatistics and Bioimaging Centre—B4, Milan, Italy. [8]Department of Oncohematology, "Annunziata" Hospital, Cosenza, Italy. [9]Medical Oncology Unit, Great Metropolitan Hospital "Riuniti" of Reggio Calabria, Reggio Calabria, Italy. [10]Institut Català d'Oncologia i Institut Josep Carreras, Badalona, Spain. [11]Servicios de Medicina Interna y Hematología, Hospital de Cabueñes, Gijón, Asturias, Spain. [12]Institut Català d'Oncologia-Hospitalet, Instituto de Investigación Biomédica de Bellvitge (IDIBELL), Barcelona, Spain. [13]Hospital Morales Meseguer, IMIB-Arrixaca, Universidad de Murcia, Murcia, Spain. [14]Hospital Universitario Puerta de Hierro,

Majadahonda, Spain. [15]Hospital Vall d'Hebron, Barcelona, Spain. [16]Hospital Universitario de Canarias, Santa Cruz de Tenerife, Spain. [17]Hospital Son Llatzer, Palma de Mallorca, Spain. [18]Hospital Clínico Lozano Blesa, Zaragoza, Spain. [19]Hospital Son Espases, Palma de Mallorca, Spain. [20]Department of Internal Medicine V, University of Heidelberg, Heidelberg, Germany. [21]Unite de Genomique du Myelome, IUC-T Oncopole, Toulouse, France. [22]Department of Hematology, ASST Spedali Civili di Brescia, Brescia, BS, Italy. [23]Cancer Research Center (IBMCC-CSIC/USAL-IBSAL), CIBER-ONC number CB16/12/00400, Salamanca, Spain. [24]Cytometry Service (NUCLEUS) and Department of Medicine,  University of Salamanca, Salamanca, Spain. [25]Hospital Clínic IDIBAPS, Barcelona, Spain. [26]These authors contributed equally: Cirino Botta, Cristina Perez.  ✉e-mail: cirino.botta@unipa.it; bpaiva@unav.es

## the Programa Para el Estudio de la Terapéutica en Hemopatías Malignas/Grupo Español de Mieloma (PETHEMA/GEM) cooperative group

Noemi Puig[3], Maria-Teresa Cedena ⓘ [4], Albert Oriol[10], Maria Esther González-Garcia[11], Anna Sureda[12], Felipe de Arriba[13], Rafael Rios Tamayo ⓘ [14], Jose-Maria Moraleda ⓘ [13], Mercedes Gironella[15], Miguel T. Hernandez[16], Joan Bargay[17], Luis Palomera[18], Albert Pérez-Montaña[19], Alberto Orfao ⓘ [23,24], Joaquin Martinez-Lopez[4], Laura Rosiñol ⓘ [25], Juan-José Lahuerta ⓘ [4], Joan Blade ⓘ [25], Maria-Victoria Mateos ⓘ [3], Jesús F. San-Miguel ⓘ [2] & Bruno Paiva ⓘ [2]✉

## the iMMunocell study group

Noemi Puig[3], Albert Oriol[10], Felipe de Arriba[13], Rafael Rios Tamayo ⓘ [14], Joan Bargay[17], Luis Palomera[18], Albert Pérez-Montaña[19], Hartmut Goldschmidt ⓘ [20], Hervé Avet-Loiseau[21], Aldo Roccaro ⓘ [22], Joaquin Martinez-Lopez[4], Maria-Victoria Mateos ⓘ [3], Jesús F. San-Miguel ⓘ [2] & Bruno Paiva ⓘ [2]✉

