## [Peer Review File · Nature Communications]

Single T cell profiles in multiple myeloma reveals dysfunction of large T cell clones and phenotypic markers of response to lenalidomide-based combinationsREVIEWER COMMENTS

Reviewer #1 (expert in TCR repertoire sequencing in cancer):

This paper characterised BM T cells from MM patients with scRNA and TCRseq. Seq data is supported by clinical data and flow cytometry analysis. Large sc sequencing dataset would also be a valuable resource to the community. Paper is succinct, well written. I would recommend revisions prior to publication.

- Is a T cell clone defined as cells with the same, unique TCRab? Not all sequenced cells will have a productive TCRab. Are cells with a sole, unique TCRa or TCRb considered a clonotype in this study? It would be important to define this in the methods

- Figure 1B – Majority of the clonotypic TCRs are found on a couple of key CD8 T cell clusters, and hardly any CD4 T cell clusters. Is this expected in BM?

- Figure 1B and C – The arbitrary definition of clonotypes is noted. If the cut-off criteria are more stringent, for example defining clonotypes as the same TCR found in 10 cells instead of 3, would the number of clonotypes change in Figure 1C? The current form of figure 1 does not give a sense if the extent of clonal expansion is different between MM, HA and MGUS/SMM.

- Figure 1E and F – See comment above regarding clonal expansion. Chao1 and ACE indices are developed for estimating richness (number of unique species) in a population. In this instance where the disease state has a reduction in richness, the important but missing data is the extent of clonotypic expansion. Renyi entropy, or an inverse shannon's index would give a sense how the abundant clones are distributed.

- Figure 2 – it is very interesting that the overall BM T cell subtypes are largely similar in proportions between different disease states. Data suggests that differences are largely in the clonotypic cells. To strengthen the argument to analyse clonotypic cells in subsequent figures, it would be helpful to demonstrate whether non-clonotypic T cells subtypes proportions were similar between disease states, similar in figure 2D and E.

- Figure 2F does not match figure legend description. There is no figure 2G in figure legend, it jumps from 2F to 2H. Regardless, the UMAP visualises expression of PD-1, LAG-3 and TIGIT on different cell subtypes. It is unclear how a P value can be derived based on a UMAP. There is some lacking data in here.

- I do not see the T cell exhausted phenotype (GZMBGZMK+LAG3+PRF1- TOX1+, line 176) that started in MGUS/SMM and peaked in MM anywhere. There is missing data here, strong emphasis for this to be displayed so that authors can make the conclusion in line 355.

- Figure 3 – Would help if more primary data from the flow cytometry analysis was displayed, especially representative FACS plots, frequency distribution of clonotypic and non-clonotypic cells in these patients. To what extent does the flow data agree with the sc seq data?

- Figure 4B and C lack detail and do not adequately show that clonality is associated with CD27. Are there meant to be discrete points in the correlation plot, and the line depicts the mean? Does the x-axis mean "number of T cell clonotypes"? It is unclear how in 4C, flow cytometry data can be correlated with number of T cell clones, which I assume is from scTCR data.

- It would be helpful if authors discuss if the differences between the two trials depicted in 5B and 5C could have contributed to differences in overall survival.

- Primary issue with Figure 5E is that CD27 was not used in this panel, but there are indeed differences in other T cell subsets in the blood before and after treatment. There is lack of explanation how this fits in with the rest of the CD27, clonotypic T cell story.

Minor comments

- Line 60 – ‘In tumors as multiple myeloma (MM)’, should be ‘In tumors such as multiple myeloma’
- Line 66 – ‘exhausted prior malignant transformation’, should be ‘exhausted prior to malignant transformation’
- Line 112 – MGUS and SMM abbreviations need to be defined. They are defined in the methods and figure legends but the results section is before the methods.
- Line 131 – though it is very likely that clonotypic cells are recognising tumour antigens, it is not supported by data. Would suggest leaving the speculative statements such as “appeared to be the result of local recognition of tumor antigens” to the discussion.
- Line 186 – MFC abbreviation needs to be defined
- Line 200 – TTP abbreviation needs to be defined.
- Line 248 – LDH needs to be defined
- Line 255 – TME needs to be defined
- Line 262 – PB needs to be defined
- Please add methodology on how different T cell subtypes were annotated via single cell transcriptome in the main methods. E.g. which reference datasets were used?
- Methods state that a mix of T cells, NK cells and monocytes were sorted for sequencing. Please clarify that CD3+ T cells were sequenced for sc?
- Figure 1D – Please add the individual data points into the graph to get a sense of T cell distribution in the patients.
- Table s10 is missing
- Methods lack description on how the list of DEG between clonal and non-clonal were calculated

Reviewer #2 (expert in single-cell RNA sequencing):

This manuscript applied multiple technologies including single-cell RNA sequencing (scRNA-seq) to analyze the T cell states and clonality of bone marrows from myeloma patients, precursors, and healthy controls. This study is merited by the scRNA-seq data of bone marrow-derived T cells, which are seldom available in the community, and the well-designed controls with myeloma. However, this manuscript has several drawbacks in data analysis.

1, Those T cell states demonstrated by the scRNA-seq data had high similarity with T cells from peripheral blood. Therefore, discrimination analysis of bone marrow-derived T cells from peripheral blood-derived T cells should be added because blood contamination may occur during the preparation of bone marrow aspirate.

2. The definition of T cell clonality is too simplified. In fact, T cell clonal expansion is highly heterogenesis. The bioinformatics tool of STARTRAC (<https://github.com/Japrin/STARTRAC>) published in Nature, 2018 will be a good tool for integrative analysis of scTCR-seq and scRNA-seq data.

3. The proposal of CD27 negative to positive ratio as a surrogate to reflect T cell clonality is less supported by data. In fact, CD27 is a marker for T cell activation only and T cell clonality can be readily measured and quantified by bulk TCR-seq. The advantage and necessity of the CD27-based metric proposed by the authors should be further justified. At least, the metric should be compared with other metrics derived from bulk TCR-seq or scTCR-seq/scRNA-seq.

Minor:

In the abstract, the patient number for scRNA-seq should be explicitly stated.

Reviewer #3 (expert in multiple myeloma):

In this study, the authors compared the clonogenic and non-clonogenic T cells throughout multiple

myeloma (MM) disease progression (precursor stage, SMM, MM) using single-cell RNA sequencing and TCR sequencing. They identified the CD27 ratio as a surrogate marker for T cell clonality and predictor for survival prior to lenalidomide therapy in MM patients. Although this kind of data analysis is novel in the field and might provide some interesting insights for future immunotherapeutic approaches, some major concerns were formulated below:

1) The abstract is a little bit misleading as it is formulated that the authors perform single-cell RNA and TCR sequencing on 1,119 patients. In the result section of Figure 1, it becomes clear that only 3 healthy, 5 MGUS/SMM precursor stages and 9 MM patients were analyzed. This number of patients is quite low and you might also expect already major differences between the MGUS and SMM stage, which are now combined for further analysis.

2) In Figure 1D, authors conclude that there is reduced diversity of clonal T cells in MGUS/SMM and MM patients. However, you can clearly observe an increase in the non-clonogenic T cell subtype as well during disease progression. Any idea why the total number of T cells increase in MM? I would also include the SD in this graph to show the inter-patient variability.

3) Based on the SD observed in healthy patients in Figure 2, it might be necessary to increase the number of analysis in this group. In Figure 2F, authors focus on the expression of immune checkpoints within clonal T cells, however from this data the inter-patient variability is unclear. Moreover, it has been concluded that there is "a trend of progressively increased expression of LAG3, PD1 and TIGIT in MM"; however first of all I don't see this trend for all immune checkpoints (rather a decrease for TIGIT) and secondly I believe this patient number is too small (and MGUS/SMM should be subdivided) to make any conclusions about the immune checkpoint expression during disease progression. A small remark, as CTLA4 is also considered as an important immune checkpoint, I would include this in the analysis.

4) At page 6, authors included the following conclusion: "results suggested the presence of an exhausted phenotype that started in MGUS/SMM and peaked in MM". I don't agree with this sentence as nothing is significantly different between healthy and MGUS/SMM in Figure 4E-F.

5) For me, it remained unclear why this specific flow cytometry panel was evaluated in Figure 3B. Previous data in Figure 3A showed increased expression of LAG3 in clonal T cells from MGUS/SMM and MM patients, while TIGIT was not mentioned at all. Why was TIGIT included in Figure 3B and not LAG3?

6) Please better describe why you focus on the selected T cell populations in Figure 3D and also include the flow cytometry gating strategy. It is not really unexpected that loss of CD27 and CD28 ('senescent phenotype') in T cell populations correlate with a poor prognosis in patients(<https://doi.org/10.1186/s13045-016-0345-3>, <https://doi.org/10.1002/cyto.a.22351>).

7) CD27 was selected in Figure 4 based on its differential expression between non-clonotypic and clonotypic T cells. This was also the case for CD28, no? Can you explain why CD27 was specifically selected for further analysis?

8) In Figure 5, the data on HLA class I/II molecules and the different T cell populations (naïve, memory etc) is a little bit confusing as it has nothing to do with the CD27 ratio.

9) It's certainly an interesting finding that the non-clonogenic and especially the clonogenic T cell population increase during disease progression. There is also some increase in LAG3 (and other immune checkpoints), however whether this is causing dysfunction/exhaustion or whether most of the T cells are still active remains unknown. It might be that the effector T cells are just hampered by immunosuppressive cell types in the microenvironment? I miss further validation of these interesting results. For example, ex vivo analysis of clonogenic T cells: are they capable of secreting cytokines, do they have effector functions, can you increase the effector function using LAG3 blocking antibodies?

Personally, I found the manuscript a combination of two different stories. On the one hand, you have the part on clonogenic/non-clonogenic T cells with the immune checkpoints that were

'increased' (not sure about that) during myelomagenesis. The increase of clonal effector T cells demonstrate that these T cells are present/active in MM; but as patients relapse, that this immune activation is unsuccessful. Whether the upregulated immune checkpoint expression on the clonal T cells is responsible for the impaired immunity in MM patients remains unknown and should be further investigated *ex vivo* to really demonstrate the link with resistance to immune checkpoint inhibitors. The second part on the CD27 ratio that could be used as a surrogate marker for T cell clonality and predicts survival after lenalidomide therapy, is interesting but not really unexpected. These data on the CD27 ratio should be expanded using other immunotherapeutic approaches (e.g., daratumumab) to elucidate whether this ratio could be a predictor for different immunotherapies.

RESPONSE TO REVIEWERS' COMMENTS

Reviewer #1 (expert in TCR repertoire sequencing in cancer):

Reviewer #1 general comment: This paper characterised BM T cells from MM patients with scRNA and TCRseq. Seq data is supported by clinical data and flow cytometry analysis. Large sc sequencing dataset would also be a valuable resource to the community. Paper is succinct, well written. I would recommend revisions prior to publication.

Answer to Reviewer #1 general comment: We thank the Reviewer for the positive opinion on our study and for the suggestions, which were addressed below and significantly improved the quality of the manuscript.

Reviewer #1 specific comment 1: Is a T cell clone defined as cells with the same, unique TCRab? Not all sequenced cells will have a productive TCRab. Are cells with a sole, unique TCRa or TCRb considered a clonotype in this study? It would be important to define this in the methods

Answer to Reviewer #1 specific comment 1: We thank the Reviewer for the comment. In the revised version of the manuscript, T cell clones were defined according to a unique TCRa and TCRb sequence (ie, TCRab). This is now defined in the Methods of the revised version of the manuscript (page 13, line 450).

Reviewer #1 specific comment 2: Figure 1B – Majority of the clonotypic TCRs are found on a couple of key CD8 T cell clusters, and hardly any CD4 T cell clusters. Is this expected in BM?

Answer to Reviewer #1 specific comment 2: The Reviewer raised an interesting question. Because these analyses are novel in precursor states and in active MM, we believe that a possible way to answer the question is by looking at the CD4/CD8 distribution of clonotypic TCRs in healthy adults. In these, the CD4/CD8 ratio of large and hyperexpanded clones was 1:1.2, whereas in MGUS/SMM and MM it was 1:1.7 and 1:2.1, respectively (Fig. S3). Thus, it appears that in normal bone marrow samples from healthy adults, large T cell clonal expansions are only slightly more frequent in CD8 T cells, and that this frequency progressively increases in MGUS/SMM and MM. Because of the Reviewer's comment, these findings are better described in the Results (page 6, lines 167-171) and the Discussion sections (page 9, lines 298-302) of the revised version of the manuscript.

Reviewer #1 specific comment 3: Figure 1B and C – The arbitrary definition of clonotypes is noted. If the cut-off criteria are more stringent, for example defining clonotypes as the same TCR found in

10 cells instead of 3, would the number of clonotypes change in Figure 1C? The current form of figure 1 does not give a sense if the extent of clonal expansion is different between MM, HA and MGUS/SMM.

Answer to Reviewer #1 specific comment 3: As the Reviewer, we also noted the lack of a cut-off consensus criterion to define clonotypes and that any number would necessary be arbitrary. Because of the comments made by all the Reviewers, we considered the possibility of systematically presenting data on all T cells rather than presenting data according to an arbitrary cut-off (i.e., comparing the phenotype of non-clonotypic vs clonotypic T cells based on a cut-off of 3 identical T cells). Accordingly, we now define T cells clones as small (clonotype range, 0% - \leq 0.01%), medium (clonotype range, 0.01% - \leq 0.1%) and large (clonotype range, 0.1% - \leq 1%), based on the percentage of T cells with identical TCRab within total T cells from each subject. This way, we could present data on all T cells based on their expansion levels in the bone marrow of healthy adults, MGUS/SMM and MM. As such, the revised version of the manuscript has been reformatted accordingly.

Regarding the specific question made by the Reviewer (if clonal expansion is different between MM, HA and MGUS/SMM), there were no significant differences in the relative distribution of small, medium and large expanded T cell clones across healthy adults, MGUS/SMM and MM (page 5, lines 155-157; Fig.1E).

Reviewer #1 specific comment 4: Figure 1E and F – See comment above regarding clonal expansion. Chao1 and ACE indices are developed for estimating richness (number of unique species) in a population. In this instance where the disease state has a reduction in richness, the important but missing data is the extent of clonotypic expansion. Renyi entropy, or an inverse shannon's index would give a sense how the abundant clones are distributed.

Answer to Reviewer #1 specific comment 4: Because of the Reviewer's comment, we have analyzed in the revised version of the manuscript the extent of clonotypic expansion using the shannon's index. There were no statistically significant differences when comparing the index score of healthy adults, MGUS/SMM and MM patients (page 5, lines 157-158; Fig. S2). We believe that this result is consistent with the lack of significant differences in the distribution of small, medium and large expanded T cell clones across healthy adults, MGUS/SMM and MM (page 5, lines 155-157; Fig.1E).

Reviewer #1 specific comment 5: Figure 2 – it is very interesting that the overall BM T cell subtypes are largely similar in proportions between different disease states. Data suggests that differences are largely in the clonotypic cells. To strengthen the argument to analyse clonotypic cells in subsequent figures, it would be helpful to

demonstrate whether non-clonotypic T cells subtypes proportions were similar between disease states, similar in figure 2D and E.

Answer to Reviewer #1 specific comment 5: We are thankful again for the Reviewer's suggestion. Because of this and the specific comment 3 made by the Reviewer, in the revised version of the manuscript we have analyzed in each cluster what was the distribution of small, medium and large T cell clones. We believe that this is a much better way of showing T cell clusters with a predominance of small clones (e.g., most CD4 subsets), and T cell clusters where a significant increment in large clones is particularly found in MM (e.g., CD8 effector GZMK and CD8 TOX+ subsets). These results are described in page 6, lines 162-171 and in the Fig. 2 of the revised version of the manuscript.

Reviewer #1 specific comment 6: Figure 2F does not match figure legend description. There is no figure 2G in figure legend; it jumps from 2F to 2H. Regardless, the UMAP visualises expression of PD-1, LAG-3 and TIGIT on different cell subtypes. It is unclear how a P value can be derived based on a UMAP. There is some lacking data in here.

Answer to Reviewer #1 specific comment 6: As noted above, Figure 2 was restructured according to the comments made by the Reviewers. UMAP of PD-1, LAG-3 and TIGIT expression are no longer shown in the revised version of the manuscript. We now show the expression of these markers specifically in large T cell clones from healthy adults, MGUS/SMM and MM patients (page 6, lines 184-186; Fig. S4).

Reviewer #1 specific comment 7: I do not see the T cell exhausted phenotype (GZMBGZMK+LAG3+PRF1- TOX1+, line 176) that started in MGUS/SMM and peaked in MM anywhere. There is missing data here, strong emphasis for this to be displayed so that authors can make the conclusion in line 355.

Answer to Reviewer #1 specific comment 7: As noted above, the interpretation of clonotypic T cells and their phenotype was updated according to the Reviewer's comments and such a statement was removed in the revised version of the manuscript. In fact, the new analyses show the increased expression of exhaustion markers in large T cell clones from MM patients, and not those with MGUS/SMM (page 6, lines 184-186; Fig. S4). As such, the revised version of the manuscript has been reformatted accordingly.

Reviewer #1 specific comment 8: Figure 3 – Would help if more primary data from the flow cytometry analysis was displayed, especially representative FACs plots, frequency distribution of clonotypic and non-clonotypic cells in these patients. To what extent does the flow data agree with the sc seq data?

Answer to Reviewer #1 specific comment 8: In accordance with the Reviewer's comment, we have added more primary data from the flow cytometry analysis in the new Fig. S7 of the revised version of the manuscript. We believe that the data in Fig. 4D and 4E shows a considerable agreement between the ratio of CD27 negative and positive T cells determined by each method and the number of T cell clones.

Reviewer #1 specific comment 9: Figure 4B and C lack detail and do not adequately show that clonality is associated with CD27. Are there meant to be discrete points in the correlation plot, and the line depicts the mean? Does the x-axis mean "number of T cell clonotypes"? It is unclear how in 4C, flow cytometry data can be correlated with number of T cell clones, which I assume is from scTCR data.

Answer to Reviewer #1 specific comment 9: We apologize for the lack of detail in the former Figs. 4B and 4C. These were meant to show the ratio between CD27 negative and positive T cells in patients with progressively increased numbers of T cell clones. Because of the Reviewer's comment, we now realize the optimal way of representing these results is by using correlation plots, which are now shown in the revised Fig. 4D and 4E.

Please note that in the Figures 4D and 4E, we correlated flow cytometry data with the number of T cell clones because both methods were simultaneously performed in bone marrow aspirates from MGUS/SMM and MM patients. Because of the Reviewer's comment, this was clarified in the revised version of the manuscript (page 7, lines 239-242).

Reviewer #1 specific comment 10: It would be helpful if authors discuss if the differences between the two trials depicted in 5B and 5C could have contributed to differences in overall survival.

Answer to Reviewer #1 specific comment 10: As the Reviewer, we also believe that differences between the two trials – namely the patients' eligibility to high-dose chemotherapy – might have contributed to the differences in survival. Another important aspect to consider is that whereas in the Claridex trial patients were treated until disease progression and, therefore, were refractory to IMiDs at the time of salvage therapy, in the GEM2012MENOS65 trial patients received fixed-duration maintenance. Thus, patients were not in an IMiD-refractory state at the time of disease progression and could have been eligible to IMiD-containing salvage regimens. Because of the Reviewer's comment, this was better discussed in the revised version of the manuscript (page 10, lines 341-346).

Reviewer #1 specific comment 11: Primary issue with Figure 5E is that CD27 was not used in this panel, but there are indeed differences in other T cell subsets in the blood before and after treatment. There is lack of explanation how this fits in with the rest of the CD27, clonotypic T cell story.

Answer to Reviewer #1 specific comment 11: We agree with the Reviewer that because CD27 was not used in the panel, it is difficult to integrate these results with previous analysis based on CD27. Because of this and the fact that the immunomodulatory effects of lenalidomide have been described elsewhere, we removed Fig. 5E from the revised version of the manuscript.

Minor comments

Reviewer #1 minor comment 1: Line 60 – ‘In tumors as multiple myeloma (MM)’, should be ‘In tumors such as multiple myeloma’

Answer to Reviewer #1 minor comment 1: We are very thankful for the Reviewer’s commitment and have corrected the sentence in the revised version of the manuscript.

Reviewer #1 minor comment 2: Line 66 – ‘exhausted prior malignant transformation’, should be ‘exhausted prior to malignant transformation’

Answer to Reviewer #1 minor comment 2: This sentence is no longer present in the revised version of the manuscript.

Reviewer #1 minor comment 3: Line 112 – MGUS and SMM abbreviations need to be defined. They are defined in the methods and figure legends but the results section is before the methods.

Answer to Reviewer #1 minor comment 3: We corrected the sentence in the revised version of the manuscript.

Reviewer #1 minor comment 4: Line 131 – though it is very likely that clonotypic cells are recognising tumour antigens, it is not supported by data. Would suggest leaving the speculative statements such as “appeared to be the result of local recognition of tumor antigens” to the discussion.

Answer to Reviewer #1 minor comment 4: We agree with the Reviewer’s comment and accordingly, the statement was moved into the Discussion section of the revised version of the manuscript (page 9, lines 294-297).

Reviewer #1 minor comment 5: Line 186 – MFC abbreviation needs to be defined

Answer to Reviewer #1 minor comment 5: We defined the multiparameter flow cytometry (MFC) abbreviation in the revised version of the manuscript (page 5, line 134).

Reviewer #1 minor comment 6: Line 200 – TTP abbreviation needs to be defined.

Answer to Reviewer #1 minor comment 6: This abbreviation is no longer present in the revised version of the manuscript.

Reviewer #1 minor comment 7: Line 248 – LDH needs to be defined

Answer to Reviewer #1 minor comment 7: We defined the lactate dehydrogenase (LDH) abbreviation in the revised version of the manuscript (page 8, line 261).

Reviewer #1 minor comment 8: Line 255 – TME needs to be defined

Answer to Reviewer #1 minor comment 8: We defined the tumor microenvironment (TME) abbreviation in the revised version of the manuscript (page 4, line 78).

Reviewer #1 minor comment 9: Line 262 – PB needs to be defined

Answer to Reviewer #1 minor comment 9: This abbreviation is no longer present in the revised version of the manuscript.

Reviewer #1 minor comment 10: Please add methodology on how different T cell subtypes were annotated via single cell transcriptome in the main methods. E.g. which reference datasets were used?

Answer to Reviewer #1 minor comment 10: The different T cell clusters were annotated considering the expression of the hallmark genes described in the new Fig. S1. This was added in the Results section of the revised version of the manuscript (page 5, lines 122-124).

Reviewer #1 minor comment 11: Methods state that a mix of T cells, NK cells and monocytes were sorted for sequencing. Please clarify that CD3+ T cells were sequenced for sc?

Answer to Reviewer #1 minor comment 11: We have clarified in the Methods section of the revised version of the manuscript that T cells were isolated according to positive expression of CD3 (page 12, line 418).

Reviewer #1 minor comment 12: Figure 1D – Please add the individual data points into the graph to get a sense of T cell distribution in the patients.

Answer to Reviewer #1 minor comment 12: This graph is no longer present in the revised version of the manuscript.

Reviewer #1 minor comment 13: Table s10 is missing

Answer to Reviewer #1 minor comment 13: We apologize. Table S10 is now shown in the revised version of the manuscript.

Reviewer #1 minor comment 14: Methods lack description on how the list of DEG between clonal and non-clonal were calculated

Answer to Reviewer #1 minor comment 14: Differentially expressed genes across T cell clones defined as small, medium and large, was calculated using the FindVariableFeatures function from the Seurat package. This was clarified in the Methods section of the revised version of the manuscript (page 12, lines 436-437).

Reviewer #2 (expert in single-cell RNA sequencing):

Reviewer #2 general comment: This manuscript applied multiple technologies including single-cell RNA sequencing (scRNA-seq) to analyze the T cell states and clonality of bone marrows from myeloma patients, precursors, and healthy controls. This study is merited by the scRNA-seq data of bone marrow-derived T cells, which are seldom available in the community, and the well-designed controls with myeloma. However, this manuscript has several drawbacks in data analysis.

Answer to Reviewer #2 general comment: We thank the Reviewer for the positive opinion on the scRNA-seq data and the well-designed controls. We are also thankful for identifying the drawbacks in data analysis, which were addressed below and significantly improved the quality of the manuscript.

Reviewer #2 specific comment 1: Those T cell states demonstrated by the scRNA-seq data had high similarity with T cells from peripheral blood. Therefore, discrimination analysis of bone marrow-derived T cells from peripheral blood-derived T cells should be added because blood contamination may occur during the preparation of bone marrow aspirate.

Answer to Reviewer #2 specific comment 1: The Reviewer is correct. Any bone marrow aspirate is affected by some level of hemodilution that is intrinsic to the collection of the specimen. This has been in fact an issue that we care deeply because of our involvement in MRD studies, in which a severely hemodiluted bone marrow aspirate can induce a false-negative result. Because of this, we systematically check the quality of each bone marrow aspirate based on the presence of bone marrow specific cell types, which are assessed using flow cytometry. These are B-cell precursors, mast cells and nucleated red blood cells. If these are absent, we consider that the level of hemodilution is very high and the sample is not representative of a bone marrow aspirate. In addition to this, we have analyzed the percentage of the three cell types in >1.400 aspirates obtained from healthy adults and patients with multiple myeloma at different stages of treatment to define median and range values (Puig, Cancers 2021), which can be used as a reference to compare with the values observed in each new sample. This way, we can estimate the degree of potential hemodilution, even in samples where B-cell precursors, mast cells and nucleated red blood cells are detectable.

Because of the Reviewer's comment, we leveraged on the flow cytometry data available from the bone marrow aspirates used for scRNA-seq, to analyze the percentages of B-cell precursors, mast cells and nucleated red blood cells, and to compare these with the median and range values from healthy adults included in the large dataset

described above (Puig, Cancers 2021). The results, which were added in Table S3 of the revised version of the manuscript, show that all samples that were analyzed in this study display percentages of B-cell precursors, mast cells and nucleated red blood cells that are indicative of representative bone marrow aspirates.

We further agree with the Reviewer that T cell states demonstrated by the scRNA-seq data had high similarity with T cells from peripheral blood. We believe that these results are somehow expected because of the typical recirculation of cells between the bone marrow and peripheral blood. To the best of our knowledge, there are no methods capable of discriminating which T cells in a bone marrow aspirate are actually resident in the marrow or blood-contaminating cells because of hemodilution. Thus, we mentioned this limitation of study in the Discussion section of the revised version of the manuscript (page 9, lines 293-294).

Reviewer #2 specific comment 2: The definition of T cell clonality is too simplified. In fact, T cell clonal expansion is highly heterogeneous. The bioinformatics tool of STARTRAC (<https://github.com/Japrin/STARTRAC>) published in Nature, 2018 will be a good tool for integrative analysis of scTCR-seq and scRNA-seq data.

Answer to Reviewer #2 specific comment 2: We are thankful to the Reviewer for indicating the bioinformatics tool STARTRAC for integrative analysis of scTCR-seq and scRNA-seq data. Accordingly, we have repeated the analyses using STARTRAC (included in the last version of scRepertoire package).

Because of this and other comments made by the Reviewer 1 regarding the arbitrary cut-off to define T-cell clonality, please note that we now define T cell clones as small (clonotype range, 0% - \leq 0.01%), medium (clonotype range, 0.01% - \leq 0.1%) and large (clonotype range, 0.1% - \leq 1%), based on the percentage of T cells with identical TCRab within total T cells from each subject. This way, we could present data on all T cells based on their expansion levels in the bone marrow of healthy adults, MGUS/SMM and MM, and avoid the use of a simplified (or arbitrary) definition of clonotypic T cells. As such, the revised version of the manuscript has been reformatted accordingly.

Reviewer #2 specific comment 3: The proposal of CD27 negative to positive ratio as a surrogate to reflect T cell clonality is less supported by data. In fact, CD27 is a marker for T cell activation only and T cell clonality can be readily measured and quantified by bulk TCR-seq. The advantage and necessity of the CD27-based metric proposed by the authors should be further justified. At least, the metric should be compared with other metrics derived from bulk TCR-seq or scTCR-seq/scRNA-seq.

Answer to Reviewer #2 specific comment 3: We agree with the Reviewer that the surrogacy between the CD27 negative to positive ratio and T cell clonality could be better supported by data other than that shown in our manuscript (mainly in Figures 4 and 5). Namely, it would be very important to compare the prognostic value of the CD27 negative to positive ratio defined by flow cytometry with that of other metrics derived from bulk TCR-seq or scTCR-seq/scRNA-seq in large series of patients. However, to our knowledge, there are currently no large cohorts (e.g., $n > 50$) with bulk TCR-seq or scTCR-seq/scRNA-seq in MM to investigate the prognostic value of T cell clonality determined using these methods. Because of the Reviewer's comment, we have acknowledged this limitation in the Discussion section of the revised version of the manuscript (page 10, lines 347-349).

Indeed, the lack of data from our and other groups on metrics derived from bulk TCR-seq or scTCR-seq/scRNA-seq in large series of patients reflects the advantage of the CD27-based metric proposed in this study. Flow cytometry is part of the diagnostic workup of MM patients and available in most Hematological clinical laboratories. Thus, the CD27 negative to positive ratio can be readily obtained and provide independent prognostic information. Because of the Reviewer's comment, the advantage of the CD27-based metric is better explained in the Discussion section of the revised version of the manuscript (page 10, lines 367-370). We further note that other markers are potentially better candidates to reflect T cell clonality (page 10, lines 353-355). Again, the reason why we focused on CD27 (that was one of the markers with different expression levels across small, medium and large T cell clones; please see Fig. 4), was the fact that it was available in the flow cytometry panel routinely used during the diagnostic workup of MM patients. As noted above, because of the Reviewer's comment, the selection of CD27 is better explained in the Discussion section of the revised version of the manuscript.

Reviewer #2 minor comment: In the abstract, the patient number for scRNA-seq should be explicitly stated.

Answer to Reviewer #2 minor comment: The patient number for scRNA-seq is now explicitly stated in the abstract of the revised version of the manuscript.

Reviewer #3 (expert in multiple myeloma):

Reviewer #3 general comment: In this study, the authors compared the clonogenic and non-clonogenic T cells throughout multiple myeloma (MM) disease progression (precursor stage, SMM, MM) using single-cell RNA sequencing and TCR sequencing. They identified the CD27 ratio as a surrogate marker for T cell clonality and predictor for survival prior to lenalidomide therapy in MM patients. Although this kind of data analysis is novel in the field and might provide some interesting insights for future immunotherapeutic approaches, some major concerns were formulated below:

Answer to Reviewer #3 general comment: We thank the Reviewer for the positive opinion on the novelty of our analysis and its potential value for future immunotherapeutic approaches. We are also thankful for the concerns raised by the Reviewer, which were addressed below and significantly improved the quality of the manuscript.

Reviewer #3 specific comment 1: The abstract is a little bit misleading as it is formulated that the authors perform single-cell RNA and TCR sequencing on 1,119 patients. In the result section of Figure 1, it becomes clear that only 3 healthy, 5 MGUS/SMM precursor stages and 9 MM patients were analyzed. This number of patients is quite low and you might also expect already major differences between the MGUS and SMM stage, which are now combined for further analysis.

Answer to Reviewer #3 specific comment 1: We apologize for the misleading numbers in the abstract. The number of patients in whom scRNA/TCR-seq was performed has been clarified in the revised version of the manuscript. Because of the Reviewer's comment, we also increased the number of cases to 4 healthy adults, 4 MGUS, 4 SMM and 10 MM.

Reviewer #3 specific comment 2: In Figure 1D, authors conclude that there is reduced diversity of clonal T cells in MGUS/SMM and MM patients. However, you can clearly observe an increase in the non-clonogenic T cell subtype as well during disease progression. Any idea why the total number of T cells increase in MM? I would also include the SD in this graph to show the inter-patient variability.

Answer to Reviewer #3 specific comment 2: Please note that in the original version of the manuscript, it was the absolute number of non-clonotypic and clonotypic T cells that was being represented in Figure 1D and not the relative distribution. Because it was the total number of T cells per group, and the number of cases in each group was different (namely higher in MM), it could be misinterpreted as an absolute increment. This was the reason why we focused on the ratio

and not on the absolute numbers of non-clonotypic and clonotypic T cells.

That notwithstanding, the Reviewer will appreciate that Figure 1 has been modified in the revised version of the manuscript. Because of the comments made by the other Reviewers regarding the cut-off to define T cell clonality and its arbitrariness, we considered the possibility of systematically presenting data of all T cells rather than presenting data according to the arbitrary cut-off of 3 identical T cells. Accordingly, we now define T cell clones as small (clonotype range, 0% - \leq 0.01%), medium (clonotype range, 0.01% - \leq 0.1%) and large (clonotype range, 0.1% - \leq 1%), based on the percentage of T cells with identical TCRab within total T cells from each subject. This way, we could present data on all T cells based on their expansion levels in the bone marrow of healthy adults, MGUS/SMM and MM. As such, the revised version of the manuscript has been reformatted accordingly.

Regarding the specific question made by the Reviewer ("Any idea why the total number of T cells increase in MM"), please note that there were no significant differences in the relative distribution of small, medium and large T cell clones across healthy adults, MGUS/SMM and MM (page 5, lines 155-157; Fig. 1E).

Reviewer #3 specific comment 3: Based on the SD observed in healthy patients in Figure 2, it might be necessary to increase the number of analysis in this group. In Figure 2F, authors focus on the expression of immune checkpoints within clonal T cells, however from this data the inter-patient variability is unclear. Moreover, it has been concluded that there is "a trend of progressively increased expression of LAG3, PD1 and TIGIT in MM"; however first of all I don't see this trend for all immune checkpoints (rather a decrease for TIGIT) and secondly I believe this patient number is too small (and MGUS/SMM should be subdivided) to make any conclusions about the immune checkpoint expression during disease progression. A small remark, as CTLA4 is also considered as an important immune checkpoint, I would include this in the analysis.

Answer to Reviewer #3 specific comment 3: As noted above in the reply to the Reviewer #3 specific comment 1, we increased the sample size to 4 HA, 4 MGUS, 4 SMM and 10 MM. Please note that in our opinion, the number of MGUS and SMM patients remains too small to subdivide both precursor conditions. Unfortunately, we could not increase the sample size more than we did because of the very high cost of the methodology in Spain.

Please note that in the new format how T cell clones are analyzed (i.e., small, medium and large clones), we observed a significant increment in the expression of immune checkpoints such as PD-1 and TIGIT in large T cell clones from MM patients vs those with MGUS/SMM and healthy adults (page 6, lines 184-186; Fig. S4). Because of the Reviewer's comment, we added CTLA4 in Figure S4, for which there

were no significant differences in the expression levels found in large T cell clones from healthy adults, MGUS/SMM and MM patients.

Reviewer #3 specific comment 4: At page 6, authors included the following conclusion: "results suggested the presence of an exhausted phenotype that started in MGUS/SMM and peaked in MM". I don't agree with this sentence as nothing is significantly different between healthy and MGUS/SMM in Figure 4E-F.

Answer to Reviewer #3 specific comment 4: As noted above, the interpretation of clonotypic T cells and their phenotype was updated according to the comments made by the other Reviewers. In fact, the new analyses point to the increased expression of exhausted markers in large T cell clones from MM patients, and not those with MGUS/SMM (page 6, lines 184-186; Fig. S4). As such, the revised version of the manuscript has been reformatted accordingly, and the statement that the Reviewer disagreed with, is no longer present in the revised version of the manuscript.

Reviewer #3 specific comment 5: For me, it remained unclear why this specific flow cytometry panel was evaluated in Figure 3B. Previous data in Figure 3A showed increased expression of LAG3 in clonal T cells from MGUS/SMM and MM patients, while TIGIT was not mentioned at all. Why was TIGIT included in Figure 3B and not LAG3?

Answer to Reviewer #3 specific comment 5: Please note that at the time we initiated the project analyzing the prognostic value of immune profiling in SMM (ie, the iMMunocell project), the novel data that is being reported in the current manuscript was unavailable. This is why we relied on the markers that were both available in the flow cytometry panels and were differentially in between non-clonotypic vs clonotypic T cells. Unfortunately, LAG3 was not included in the flow cytometry panel and that is the reason why it could not be analyzed. We agree with the limitations of this analysis and because of the Reviewer's comment, it was removed from the revised version of the manuscript.

Reviewer #3 specific comment 6: Please better describe why you focus on the selected T cell populations in Figure 3D and also include the flow cytometry gating strategy. It is not really unexpected that loss of CD27 and CD28 ('senescent phenotype') in T cell populations correlate with a poor prognosis in patients (<https://doi.org/10.1186/s13045-016-0345-3>, <https://doi.org/10.1002/cyto.a.22351>).

Answer to Reviewer #3 specific comment 6: As noted in the answer to the previous comment, the focus on the selected T cell populations in Figure 3D was both guided by the differentially

expressed genes (coding for CD markers) in non-clonotypic vs clonotypic T cells according to the scRNA/TCR-seq data, and the CD markers available in the flow cytometry panels being analyzed in the SMM cohort. As noted above, this section was removed from the revised version of the manuscript due to the limitations highlighted by the Reviewer.

Reviewer #3 specific comment 7: CD27 was selected in Figure 4 based on its differential expression between non-clonotypic and clonotypic T cells. This was also the case for CD28, no? Can you explain why CD27 was specifically selected for further analysis?

Answer to Reviewer #3 specific comment 7: For the exact same reason described in the answer to the Reviewer's comments 5 and 6. Please note that CD27 (and not CD28) is part of the flow cytometry panel we use for the screening of MM patients at diagnosis. Thus, the reason why we focused on CD27 (that was one of the markers with different expression levels across small, medium and large expanded T cells; please see Fig. 4), was the fact that it was available in the flow cytometry panel routinely used during the diagnostic workup of MM patients. Because of the Reviewer's comment, the selection of CD27 is better explained in the Discussion section of the revised version of the manuscript (page 10, lines 355-357).

Reviewer #3 specific comment 8: In Figure 5, the data on HLA class I/II molecules and the different T cell populations (naïve, memory etc) is a little bit confusing as it has nothing to do with the CD27 ratio.

Answer to Reviewer #3 specific comment 8: We are thankful for the Reviewer's comment and in accordance, we have removed both panels and corresponding data in the revised version of the manuscript.

Reviewer #3 specific comment 9: It's certainly an interesting finding that the non-clonogenic and especially the clonogenic T cell population increase during disease progression. There is also some increase in LAG3 (and other immune checkpoints), however whether this is causing dysfunction/exhaustion or whether most of the T cells are still active remains unknown. It might be that the effector T cells are just hampered by immunosuppressive cell types in the microenvironment? I miss further validation of these interesting results. For example, ex vivo analysis of clonogenic T cells: are they capable of secreting cytokines, do they have effector functions, can you increase the effector function using LAG3 blocking antibodies?

Answer to Reviewer #3 specific comment 9: We thank the Reviewer for the positive opinion about these findings. We have thought carefully on how to validate these results, and envisioned that one approach was to evaluate the efficacy of blocking antibodies

against LAG3 and other immune checkpoints, in the recent mouse models of MM progression developed by our colleague Dr. Martinez-Climent (Larrayoz M, et al. Nature Medicine 2023 <https://pubmed.ncbi.nlm.nih.gov/36928817/>).

In this new section of the revised version of the manuscript (please see the Results section “ICB combination therapy tailored to the phenotype of large T cell clones” and Figure 3), we first characterized T cell phenotypes and TCR predominance in mice as we did in humans. Notwithstanding some expected differences, we found many similarities such as the T cell cluster composition in large T cell clones from control, MGUS and MM mice. Regarding the expression of immune checkpoints in large T cell clones, we observed increased expression of TIGIT and PD1 in MM vs MGUS and control mice. Because of these findings and the Reviewer’s comment, we added in the revised version of the manuscript the experiments of the combined administration of anti-PD1 plus anti-LAG3 or anti-TIGIT (page 7, lines 214-224; Fig. 3E). Interestingly, none of the ICB used in monotherapy prolonged survival; by contrast, the co-administration of anti-PD1 plus anti-LAG3, or anti-PD1 plus anti-TIGIT, resulted in longer overall survival. We believe that these results suggest increased effector function using immune checkpoint blocking antibodies as the Reviewer wisely anticipated.

Reviewer #3 specific comment 10: Personally, I found the manuscript a combination of two different stories. On the one hand, you have the part on clonogenic/non-clonogenic T cells with the immune checkpoints that were ‘increased’ (not sure about that) during myelomagenesis. The increase of clonal effector T cells demonstrate that these T cells are present/active in MM; but as patients relapse, that this immune activation is unsuccessful. Whether the upregulated immune checkpoint expression on the clonal T cells is responsible for the impaired immunity in MM patients remains unknown and should be further investigated ex vivo to really demonstrate the link with resistance to immune checkpoint inhibitors. The second part on the CD27 ratio that could be used as a surrogate marker for T cell clonality and predicts survival after lenalidomide therapy, is interesting but not really unexpected. These data on the CD27 ratio should be expanded using other immunotherapeutic approaches (e.g., daratumumab) to elucidate whether this ratio could be a predictor for different immunotherapies.

Answer to Reviewer #3 specific comment 10: Because of the comments made by the three Reviewers, we believe that the quality of the manuscript was considerably improved. Furthermore, the relationship between T cell phenotypes and clonality across disease progression is described with greater clarity.

As noted above, the possible link between the upregulated immune checkpoint expression on large T cell clones and their function

was investigated in experimental MM models. Indeed, because of the Reviewer's comment about the lack of functional studies, we have performed additional *in vitro* experiments to gain further insight into a putative association between large T cell clones and their re-activation upon exposure to lenalidomide. Accordingly, we cultured whole BM aspirates from MM patients (n = 3) in a 3D organoid during five days, and treated with 1 μ M lenalidomide +/- an anti-HLA antibody to block TCR-MHC interactions. The significant tumor cell killing induced by lenalidomide was nearly abrogated in the presence of HLA blocking. We believe that these new data reinforce the possible association between the prognostic value of the CD27 ratio and the re-activation of large T cell clones upon lenalidomide-based combination therapy. These results were added in the revised version of the manuscript (page 8, lines 269-277; Fig. 5E). This way, we hope that the revised version of the manuscript does not appear to be the combination of two different stories.

Regarding the final comment made by the Reviewer about expanding the data on the CD27 ratio in patients treated with other immunotherapeutic approaches (e.g., daratumumab); please note that unfortunately, we do not have mature follow-up in large series of patients with data on the CD27 ratio treated in newer clinical trials. Because of the Reviewer's comment, we have acknowledged this limitation in the Discussion section of the revised version of the manuscript (page 10, lines 369-370).

REVIEWER COMMENTS

Reviewer #1 (TCR repertoire sequencing in cancer):

I thank the authors for the additional experiments and edits to the paper. My comments pertain to the T cell receptor sequencing and analysis.

The primary concern I have is the distribution of cells in the patient population is not clearly delineated. For example, the dysfunctional T cell population (CD8+TOX+) was present in some (mostly in one patient), but not all the MM patients.

It would be also be important to present whether PD-1 and TIGIT expressing large clones are in most MM patients, or just a few.

Other major comment is that statistical support for figure 2B and 3B is required to claim that some populations are enriched in large clones but not small clones.

Minor comments:

Line 94 'where' instead of 'while';

Line 156 – Fig 1F called out before E

Line 165 – Tregs, not Treg

Line 219 Pd1, Lag3 and Tigit not capitalised

Figure 3C has an extra box that covers CD4+ EM

Personally I felt that the addition of a murine model changed the story of the manuscript significantly. I will defer to reviewer 3 for their expert opinion on the relevance of the model.

Reviewer #2 (expert in single-cell RNA sequencing):

The authors have addressed all my concerns by adding new data or discussion. The manuscript is now acceptable for publication. The rich data presented in this study is very useful for understanding the cellular composition of human bone marrow.

Reviewer #3 (expert in multiple myeloma):

The comparison of clonogenic and non-clonogenic T cells throughout multiple myeloma disease progression (precursor stage, SMM, MM) using single-cell RNA sequencing and TCR-sequencing is of high interest for the MM community. Although the number of analyzed patient samples remains limited, reviewers' comments have been sufficiently addressed by the authors and additional experiments significantly improved the quality of the manuscript.

RESPONSE TO REVIEWERS' COMMENTS

Reviewer #1 (TCR repertoire sequencing in cancer):

Reviewer #1 general comment: I thank the authors for the additional experiments and edits to the paper. My comments pertain to the T cell receptor sequencing and analysis.

Answer to Reviewer #1 general comment: We thank the Reviewer for the positive opinion about the additional experiments and edits to the paper. We are also thankful for the thorough Review and the additional comments, which were addressed below and improved the quality of the manuscript.

Reviewer #1 specific comment 1: The primary concern I have is the distribution of cells in the patient population is not clearly delineated. For example, the dysfunctional T cell population (CD8+TOX+) was present in some (mostly in one patient), but not all the MM patients.

Answer to Reviewer #1 specific comment 1: Because of the Reviewer' comment, the revised version of the manuscript includes the original Table S4 (with the distribution of cell clusters in each case) and a new Fig. S2 with the distribution of each cell cluster in healthy adults, MGUS/SMM and MM patients. Also because of the Reviewer' comment, we added in the main text of the revised version of the manuscript that the dysfunctional T cell population (CD8+TOX+) was present in only a few (mostly in one) MM patient (page 5, line 142).

Reviewer #1 specific comment 2: It would be also be important to present whether PD-1 and TIGIT expressing large clones are in most MM patients, or just a few.

Answer to Reviewer #1 comment 2: Because of the Reviewer' comment, we have modified the Fig. S5 so that the expression of PD-1, TIGIT and the other markers in large clones can be analyzed in each subject. Of note, PD-1 and TIGIT are generally more expressed in large clones from most MM patients.

Reviewer #1 specific comment 3: Other major comment is that statistical support for figure 2B and 3C is required to claim that some populations are enriched in large clones but not small clones.

Answer to Reviewer #1 comment 3: Because of the Reviewer' comment, we added statistical support in Figures 2B and 3C. We also made minor modifications in the main text of the revised version of the manuscript to reflect the statistical significance of differences in large clones (page 6, lines 173-179 and lines 206-208).

Minor comments:

Reviewer #1 minor comment 1: Line 94 'where' instead of 'while;

Reviewer #1 minor comment 2: Line 156 – Fig 1F called out before E

Reviewer #1 minor comment 3: Line 165 – Tregs, not Treg

Reviewer #1 minor comment 4: Line 219 Pd1, Lag3 and Tigit not capitalized

Reviewer #1 minor comment 5: Figure 3C has an extra box that covers CD4+ EM

Answer to Reviewer #1 minor comments 1-5: We are very thankful for the Reviewer' commitment to improve the quality of the manuscript. We have corrected the revised version accordingly.

Reviewer #1 minor comment 6: Personally I felt that the addition of a murine model changed the story of the manuscript significantly. I will defer to reviewer 3 for their expert opinion on the relevance of the model.

Answer to Reviewer #1 minor comments 6: We than the Reviewer for the comment. The Reviewer 3 commented below that the "reviewers' comments have been sufficiently addressed by the authors and additional experiments significantly improved the quality of the manuscript."

Reviewer #2 (expert in single-cell RNA sequencing):

Reviewer #2 general comment: The authors have addressed all my concerns by adding new data or discussion. The manuscript is now acceptable for publication. The rich data presented in this study is very useful for understanding the cellular composition of human bone marrow.

Answer to Reviewer #2 general comment: We thank the Reviewer for the positive opinion on the rich data presented in this study and its utility for understanding the cellular composition of human bone marrow.

Reviewer #3 (expert in multiple myeloma):

Reviewer #3 general comment: The comparison of clonogenic and non-clonogenic T cells throughout multiple myeloma disease progression (precursor stage, SMM, MM) using single-cell RNA sequencing and TCR-sequencing is of high interest for the MM community. Although the number of analyzed patient samples remains limited, reviewers' comments have been sufficiently addressed by the authors and additional experiments significantly improved the quality of the manuscript.

Answer to Reviewer #3 general comment: We thank the Reviewer for acknowledging the effort of the additional samples and experiments, and how these significantly improved the quality of the manuscript.

REVIEWERS' COMMENTS

Reviewer #1 (expert in TCR repertoire sequencing):

Thank you for addressing the queries and comments, the data will be a valuable resource to the community.

RESPONSE TO REVIEWERS' COMMENTS

Reviewer #1 (TCR repertoire sequencing in cancer):

Reviewer #1 general comment: Thank you for addressing the queries and comments, the data will be a valuable resource to the community.

Answer to Reviewer #1 general comment: We thank the Reviewer for the positive opinion about the additional experiments and edits to the paper. We are also thankful for the thorough Review that improved the quality of the manuscript.